# ECOSMO II(CHL): a marine biogeochemical model for the North Atlantic and the Arctic

Veli Çağlar Yumruktepe[1], Annette Samuelsen[1], Ute Daewel[2]

[1] Nansen Environmental and Remote Sensing Center, Jahnebakken 3, N-5007, Bergen, Norway

[2] Helmholtz-Zentrum Hereon, Institute for Coastal Systems – Analysis and Modelling, Max-Planck-Str. 1, Geesthacht, Germany

*Correspondence to*: Veli Çağlar Yumruktepe (caglar.yumruktepe@nersc.no)

**Abstract.** ECOSMO II is a fully coupled bio-physical model of 3d-hydrodynamics with an intermediate complexity N(utrient) P(hytoplankton) Z(ooplankton) D(etritus) type biology including sediment-water column exchange processes originally

formulated for the North Sea and Baltic Sea. Here we present an updated version of the model incorporating chlorophyll *a* as a prognostic state variable: ECOSMO II(CHL). The version presented here is online coupled to the HYCOM ocean model. The model is intended to be used for regional configurations for the North Atlantic and the Arctic incorporating coarse to high spatial resolutions for hind-casting and operational purposes. We provide the full descriptions of the changes in ECOSMO II(CHL) from ECOSMO II and provide the evaluation for the inorganic nutrients and chlorophyll *a* variables, present the

modelled biogeochemistry of the Nordic Seas and the Artic and experiment on various parameterization sets as use cases targeting chlorophyll *a* dynamics. We document the performance of each parameter set objectively analysing the experiments against in situ, satellite and climatology data. The model evaluations for each experiment demonstrated that the simulations are consistent with the large-scale climatological nutrient setting and are capable of representing regional and seasonal changes. Explicitly resolving chlorophyll *a* allows for more dynamic seasonal and vertical variations in phytoplankton biomass to

chlorophyll *a* ratio and improves model chlorophyll *a* performance near the surface. Through experimenting the model performance, we document the general biogeochemisty of the Nordic Seas and the Arctic. The Norwegian and Barents Seas primary production show distinct seasonal patterns with a pronounced spring bloom dominated by diatoms and low biomass during winter months. The Norwegian Sea annual primary production is around double that of the Barents Sea while also having an earlier spring bloom.

## 1 Introduction

Operational ocean forecasting and reanalysis systems that integrate in situ measurements, remote sensing observations, modelling and data-assimilation are fundamental tools for understanding the variability and dynamics of the physical and biogeochemical ocean state. Such systems are also essential for a better and more sustainable management of the oceans and marine ecosystems, supporting the development and understanding of human activities and the blue economy (von

Schuckmann et al., 2016). In this context, the presentation of the underlying science, continuous evaluation and development of the forecast systems are required to provide the best possible forecast and reanalysis.

The presented model version, ECOSMO II(CHL), is adapted from the biogeochemical model ECOSMO (Schrum et al., 2006), later ECOSMO II (Daewel and Schrum, 2013; DS2013), and is currently used as the marine biogeochemical model for

operational forecasts (https://doi.org/10.48670/moi-00003) of the Arctic Ocean (ARC MFC – Arctic Marine Forecasting Centre) under the umbrella of CMEMS (The European Copernicus Marine Environment Monitoring Service; marine.copernicus.eu). The biogeochemical forecast ECOSMO II(CHL) has been operational since April 2017 and the daily values of the selected variables can be retrieved from the CMEMS database. While based on the ECOSMO II version presented in DS2013, the transfer of the model to a different circulation model, region, and model resolution necessitated an adjustment

of model parameterizations and additional functionalities, which in turn required a series of new sensitivity tests.

ECOSMO II is an intermediate complexity nutrient-phytoplankton-zooplankton-detritus (NPZD) type model describing the trophic interactions between three phytoplankton and two zooplankton components. It was shown to successfully simulate the seasonal and inter-annual ecosystem variability of primary and secondary production in the North- and Baltic Sea (Daewel

and Schrum, 2013). In the framework of the ARC MFC forecasting system which covers the northern part of the Atlantic Ocean and the Arctic, its application and scientific scope was shifted to be used for the open ocean and sea-ice covered domains.  Furthermore, when moving from one circulation model to another, biogeochemical models will behave differently as a result of differences in the physical model (Skogen and Moll, 2005). Both these changes require adjustments to the model formulation and parameters to give good result in the focus regions. ECOSMO II(CHL) most notably introduces chlorophyll

*a* as a prognostic variable. Allowing a flexible chlorophyll-to-carbon ratio is more realistic and has been shown to be more stable when chlorophyll is assimilated (Ciavatta et al., 2011). This addition allows the direct assimilation of ocean color observations into the forecasting and reanalysis systems. The description of the model changes, added components and the evaluation of the ECOSMO II(CHL) results within the North Atlantic and Arctic form the main content of this paper.

The North Atlantic above 60°N, the focus in this paper, is a typical spring-bloom system (Longhurst, 1998; Rey, 2004). During winter, strong winds and cooling mix the water column several hundred meters and brings up nutrients-rich waters (Nilsen and Falck, 2006). Once the water column stratifies enough for the bloom to start, the diatoms dominate the system. When silicate is depleted, the smaller flagellates and dinoflagellates dominate the phytoplankton community (Rey, 2004). Sporadically there are also extensive coccolithophores blooms covering large areas (Baumann et al., 2000). The main species of mesozooplankton

in this area, *Calanus finmarchicus*, overwinters at depth (Melle et al., 2004) and ascend to the surface at the onset of spring, therefore there is already some zooplankton biomass present at the time of the start of the spring. There is also a fall bloom present as seen from satellite observations. The areas closer to the Arctic, being covered by sea ice, have different dynamics. In sea ice covered regions, small blooms can occur in leads and under thin ice but the main bloom commences as the ice

retracts (Dalpadado et al., 2020; Dong et al., 2020; Polyakov et al., 2020). Here, sea ice algae will make up some of the primary production (Gradinger, 2009) and other mesozooplankton, such as *Calanus glacialis*, specialized to the sea ice environment (Melle and Skjoldal, 1998), are also important. Close to sea ice and in coastal regions, early stratification can occur when sea or land ice melts resulting in a seasonal halocline. Water masses in the eastern part of the basin are relatively warm, saline water characterizing the North Atlantic Current (Orvik et al., 2001), while the western part of the basin has colder and fresher water masses with an Arctic or mixed origin (Fröb et al., 2018 Yashayaev et al., 2007).

Our main objective with this paper is to provide the description of the latest updates in ECOSMO II(CHL) and its coupling to HYCOM. We will particularly focus on the description of the prognostic chlorophyll *a* formulation. We present the results from three experiments using ECOSMO II(CHL) adopting different parameter sets from DS2013 (the original parameter set tuned for the North and Baltic Seas), CMEMS Artic operational model prior to June 2021 and the current Arctic operational model parameterization. We applied these 3 parameter sets on a model setup with a coarser grid than used for the operational simulations in order to allow for 2-decade long simulations for each case. To document the performance of each of these parameter sets for the users of ECOSMO, we present a detailed objective analysis of the lower trophic level dynamics for the North Atlantic and the Arctic Ocean against local in situ observations, gridded climatology of nutrients and satellite data in Sect. 4. Following the evaluation we provide information on integrated quantities, such as annual primary production, inter-annual variability in phytoplankton production and seasonal succession of plankton functional types as a reference for the Nordic Seas and the Arctic. We will finalize by commenting on the future updates and implementations of ECOSMO II(CHL).

## 2 The HYCOM-ECOSMO II(CHL) model

HYCOM-ECOSMO II(CHL) is a coupled physical-biological model (Fig. 1) where ocean physics are represented by the Hybrid Coordinate Ocean Model (HYCOM: Bleck, 2002) and the lower trophic marine biogeochemistry is resolved by ECOSMO II (Daewel and Schrum, 2013). The models are coupled online and the transport (advection and mixing) of biological state variables is handled as part of HYCOM's own native tracer-transport routines, thus both the physical and biological components use the same time stepping (20 minutes). The model is one-way coupled and biology does not affect model physics. HYCOM, as a hybrid vertical coordinate model, can optionally combine the depth-level (z-level), topography following and density-following (isopycnal) coordinates. In this study, we set vertical levels as the combination of z-level for the upper ocean and the mixed layer and isopycnal layers below. The upper 5 layers are always kept in z-levels ensuring a minimum vertical resolution which is important to resolve the light gradient in the upper ocean and thus representing the vertical variation in phytoplankton growth in a realistic manner. Isopycnal layers in the deep facilitates a good conservation of water-masses and tracer distributions.

ECOSMO II(CHL) is an intermediate-complexity lower trophic level biogeochemical model which distinguishes four inorganic nutrients (nitrate, ammonium, phosphate and silicate) utilized by three types of phytoplankton (diatoms, flagellates and cyanobacteria). In this study, cyanobacteria are turned off, as they were parameterized to grow below a certain salinity threshold which was intended to represent the cyanobacteria in the Baltic Sea (Daewel and Schrum, 2013). Our area of concern is the high latitudes, specifically the area north of 60°N, thus the use of cyanobacteria falls short as a significant phytoplankton community for the region. Two types of zooplankton (micro- and meso-size classes) are parameterized based on their feeding preferences as herbivorous and omnivorous zooplankton and, as additional organic components, dissolved (DOM) and particulate (detritus) organic matter are included in the model. The model uses the molar Redfield ratio between C:N:Si:P components (106 : 6.625 : 6.625 : 1), and nutrients are tracked both in the water column and in the single sediment layer.

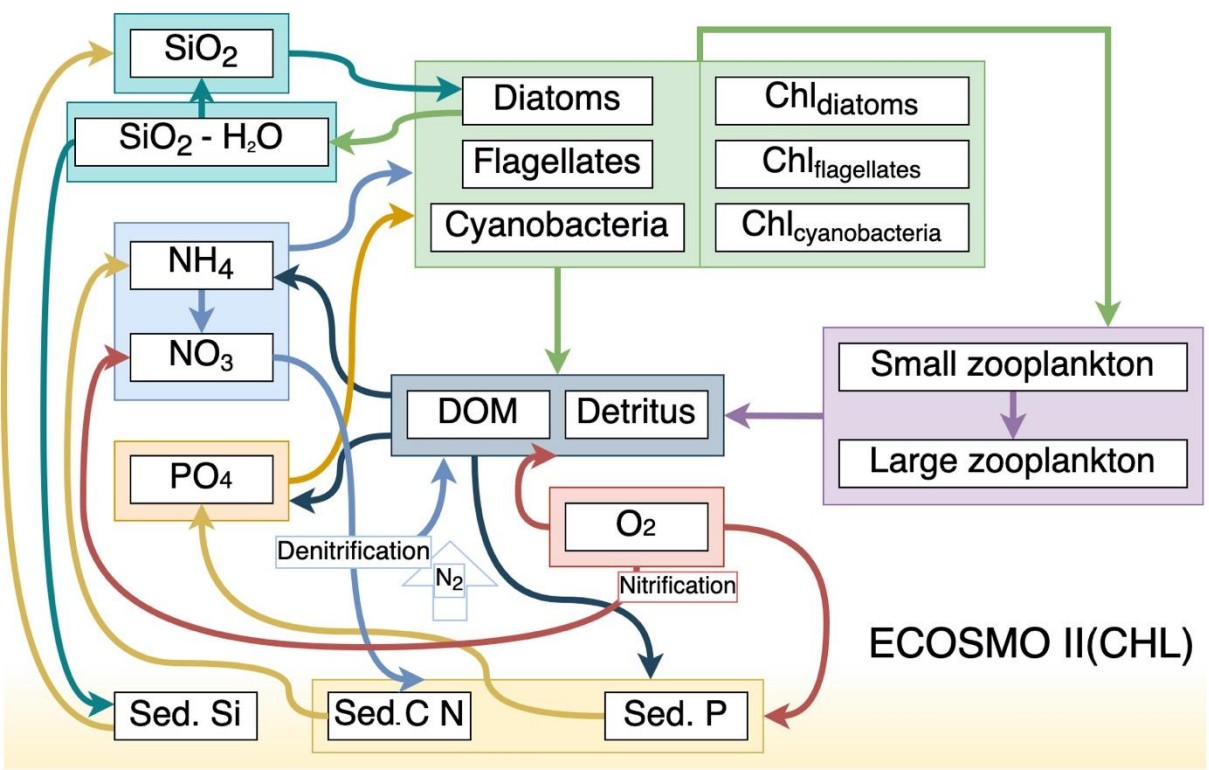

**Fig. 1: Schematic diagram of biochemical interactions in ECOSMO II. (DOM: dissolved organic matter; Chl- prefixes stand for phytoplankton type specific chlorophyll _a_ content; Sed. denote sediment pool with silicate, phosphorus and nitrate content.)**

The full description of ECOSMO II is given in Daewel and Schrum (2013) (DS2013). In the following we provide a description of differences in the biogeochemical formulations in ECOSMO II(CHL) compared to DS2013. The most notable addition to DS2013 is the prognostic chlorophyll _a_ for each phytoplankton type. The biological interaction ($R_{chl_j}$) term of the introduced chlorophyll _a_ for $P_1$ and $P_2$ (diatoms and flagellates respectively) is in

similar fashion to that of $R_{P_j}$ in DS2013, and the source terms are modified by the photoacclimation factor ($\rho_{chl_j}$) which accounts for the variation in chlorophyll-to-biomass ratio resulting in increased chlorophyll production under low light conditions (Geider et al., 1997), hence:

$$R_{chl_j} = \rho_{chl_j}\sigma_j\phi_{P_j}C_{P_j} - \sum_{i=1}^{2} G_i(P_j)C_{Z_i}\frac{Chl_{P_j}}{C_{P_j}} - m_{P_j}Chl_{P_j} \tag{1}$$

where,

$$\rho_{chl_j} = \frac{\theta_P^{max}\phi_{P_j}C_{P_j}}{\alpha_{P_j}I(x,y,z,t)\,Chl} \tag{2}$$

$$\phi_{P_j} = \min\left(\alpha(I),\beta_N,\beta_P,\beta_{Si}\right) \tag{3}$$

$$\alpha(I) = \tanh\left(\varphi I(x,y,z,t)\right) \tag{4}$$

$$\beta_N = \beta_{NH_4} + \beta_{NO_3} \tag{5}$$

$$\beta_{NH_4} = NH_4/(NH_4 + r_{NH_4}) \tag{6}$$

$$\beta_{NO_3} = (NO_3/(NO_3 + r_{NO_3}))\exp\left(-\gamma NH_4\right) \tag{7}$$

$$\beta_{PO_4} = PO_4/(PO_4 + r_{PO_4}) \tag{8}$$

$$\beta_{Si} = Si/(Si + r_{Si}) \tag{9}$$

$$G_i(P_j) = \sigma_{i,P_j}\frac{a_{i,P_j}C_{P_j}}{r_i + F_i} \tag{10}$$

$$F_i = \sum_{j=1}^{2} a_{i,P_j}C_{P_j} \tag{11}$$

with j = 1,2 denote the specific phytoplankton types and i = 1,2 the specific zooplankton types. C denote carbon concentration specific to P (phytoplankton) and Z (zooplankton) in mg m$^{-3}$, while Chl denote chlorophyll *a* concentration in mg m$^{-3}$. Photosynthetically active radiation (PAR) is given as I(x,y,z,t). DS2013 give $\sigma_j$, $\phi_{P_j}$, $\varphi$, $r_{NH_4,NO_3,PO_4,Si}$, $\gamma$, $G_i$ and $m_{P_j}$ as the phytoplankton maximum growth rate, growth limitation, photosynthesis efficiency parameter, nutrient-specific half saturation constant, NH$_4$ inhibition parameter, zooplankton grazing rates and mortality rates respectively. $\sigma_{i,P_j}$ denotes zooplankton specific grazing rate with $a_{i,P_j}$ and $r_i$ representing food preference coefficient and half saturation constant respectively where $F_i$ denote the total available food for the individual zooplankton. Silicate is not included in flagellate equations. Maximum Chl-to-C ratio ($\theta_P^{max}$) is taken from Bagniewski et al. (2011), where they have tuned those parameters for the region south of Iceland. We note that their parameterization is N-based, while ECOSMO II(CHL) uses C-based parameters, thus we applied the conversion following the C:N Redfield ratio of 6.625 resulting in flagellates and diatoms to have 0.048 and 0.037 mgChl

mgC$^{-1}$ respectively. In relation to the addition of a prognostic chlorophyll $a$ state variable, photosynthetically active radiation I(x,y,z,t) at depth undergoing attenuation was modified to have chlorophyll $a$ in the exponential term:

$$I(x, y, z, t) = \frac{I_s(x,y)}{2} exp \left( -k_w z - k_{Chl} \int_z^0 \sum_{j=1}^2 Chl_{P_j} \partial z \right) \tag{12}$$

where $I_s(x, y)$ is the surface net solar radiation (W m$^{-2}$) converted to PAR, and $x, y$ identifies the models horizontal grid points, with $z$ the water depth in meters. $k_w$ and $k_{Chl}$ are light extinction due to water (m$^{-1}$) and chlorophyll $a$ concentration (m$^2$ mgChl$^{-1}$) respectively.

In addition to prognostic chlorophyll $a$ state variables, phytoplankton and zooplankton loss terms now have an on/off switch regulated by a minimum concentration criterion preventing them from decreasing to very low concentrations. This allows them to recover and quickly respond to suitable growth conditions experienced in spring. The switch is applied to mortality and grazing terms for phytoplankton and chlorophyll $a$, and to mortality terms for zooplankton. The minimum concentration at

150 which the loss terms are switched off are 0.1, 0.005 and 0.01 mgC m$^{-3}$ for phytoplankton, chlorophyll $a$ and zooplankton respectively.

**3 Model setup and evaluation framework**

Model simulations are configured on a relatively coarse grid that varies between 30 and 70 km where the highest resolutions are located in the mid-North Atlantic (Fig. 2). Although, having finer resolution was previously shown to better represent

nutrient dynamics for our domain (Samuelsen et al., 2015), the main purpose of our study is to introduce the required model structure for the North Atlantic/Arctic region and the experiments, which requires numerous tests and simulations in parallel. Therefore, we concluded that having a relatively coarse grid size fits better for our purposes.

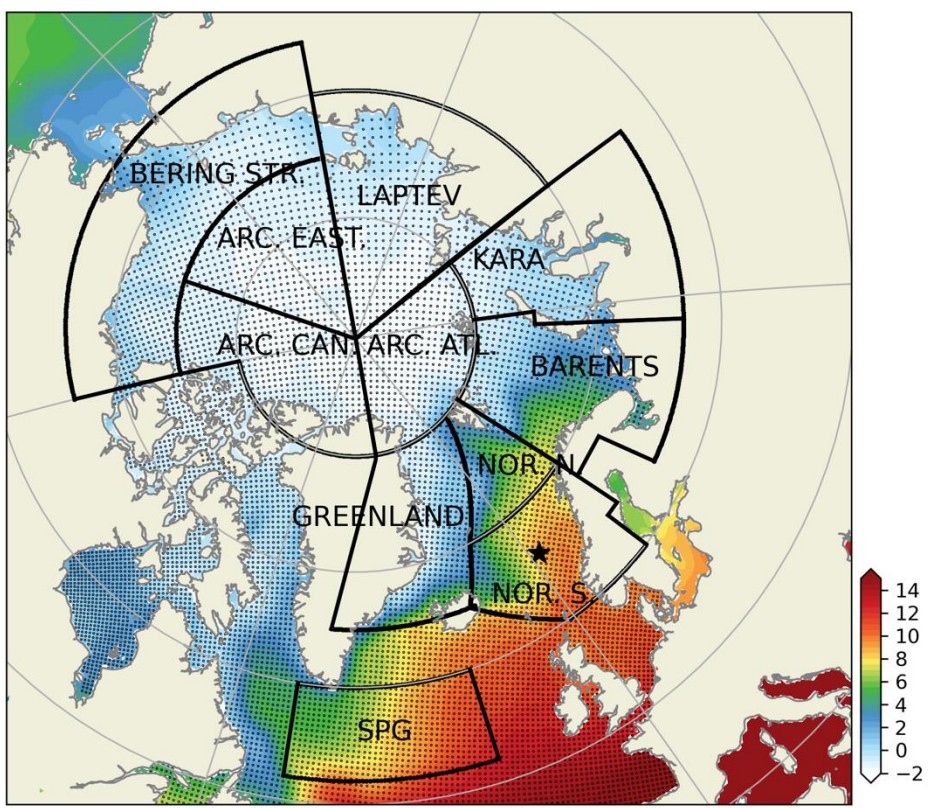

**Fig. 2: Subdivision of model domain in prescribed geographical subdomains used for model quality assessments. The subdomains are as follows: Norwegian Sea South (NOR. S.), Norwegian Sea North (NOR. N.), Barents Sea (BARENTS), Kara Sea (KARA), Laptev Sea (LAPTEV), Bering Strait (BERING STR.), Arctic-Canada (ARC. CAN.), Arctic-East (ARC. EAST), Arctic-Atlantic (ARC. ATL.), Greenland Sea (GREENLAND) and the Subpolar Gyre (SPG). The black points in the oceanic regions denote the model grid coordinates. The coordinates for the Station-M time-series station location is depicted with the star. While the model domain extends down to the equatorial regions, the figure focuses on the area of interest. Note that the BERING STR. subdomain is within the effective area of the open boundary conditions thus is relaxed to climatology. WOA18 1981 – 2010 annual surface temperature climatology (Boyer *et al*., 2018) is depicted with the coloured shades.**

Data for atmospheric forcing is retrieved from ECMWF ERA-Interim reanalysis with 6-hour resolution (Dee et al., 2011). The variables used to force the ocean model are 10m winds, air temperature at 2 meters, dew-point temperature at 2 meters, cloud coverage and total precipitation for the physical model and surface net solar radiation for the biogeochemical model. River runoff is modelled using a hydrological model, TRIP (Oki et al., 2009), resulting in a monthly climatology dataset, so the river runoff does not include any interannual variability. River runoff affects only salinity. Nutrient loads from the rivers are derived from the modelled dataset, GlobalNEWS (Mayorga et al., 2010; Seitzinger et al., 2010), which include nitrate, phosphate and silicate. Nutrient loads were scaled by the TRIP runoff volume resulting in monthly climatology loads.

The model physics was initialized in 1989 from a spin-up simulation that started in 1948 forced by the ECHAM6 atmospheric simulation (Schubert-Frisius and Feser, 2015). The biogeochemical model used inorganic nutrients (nitrate, phosphate and silicate) from the World Ocean Atlas 2013 (Garcia et al., 2013) monthly climatology as the initial conditions; the biomass concentrations were initialized with uniform, low values. The same climatology was used for the relaxation of temperature,
salinity, nitrate, silicate, phosphate and oxygen at the open boundaries. The simulation was conducted until the end of 2010. The results are evaluated starting with the year 1991.

**Table 1: Parameters that were modified between different experiments**

| | Model experiments | | |
|---|---|---|---|
| | EXP1 | EXP2 | EXP3 |
| Diatom maximum growth rate ($\sigma_{P_1}$) (1 day$^{-1}$) | 1.3 | 1.95 | 1.75 |
| Flagellate maximum growth rate ($\sigma_{P_2}$) (1 day$^{-1}$) | 1.1 | 1.65 | 1.45 |
| Photosynthesis efficiency ($\varphi$) (m$^2$ W$^{-1}$) | 0.03 | 0.01 | 0.012 |
| Mesozooplankton grazing rate on phytoplankton ($\sigma_{i,P_{j,}}$) (1 day$^{-1}$) | 0.8 | 1.2 | 1.2 |
| Mesozooplankton grazing rate on microzooplankton ($\sigma_{i,Z_{micro,}}$) (1 day$^{-1}$) | 0.5 | 0.75 | 0.75 |
| Microzooplankton grazing rate on phytoplankton ($\sigma_{i,P_{j,}}$) (1 day$^{-1}$) | 1.0 | 1.5 | 1.5 |

In this study, we employ 3 set of simulations (EXP1, EXP2 and EXP3) that use different phytoplankton growth rates, photosynthesis efficiency, and zooplankton mortality rates (Table 1). EXP1 uses the DS2013 parameter set which was used for the North Sea and the Baltic Sea. Additionally, we introduce EXP2 (uses the parameter set for the operational forecast model for ARC MFC prior to July 2021) and EXP3 (uses the parameter set for the operational forecast model for ARC MFC currently online following July 2021). Since these parameter sets represent active use cases, the objective analysis of these
experiments in the following sections provide the users of ECOSMO a reference on how the model performs with different setups and longer time-scales. For the purpose of comparing these parameters, EXP2 and EXP3 can be considered as part of the same group against EXP1 such that in both EXP2 and EXP3, phytoplankton growth rates are set higher compared to EXP1. The reasoning behind this increase is a response to deep winter convective mixing and resulting light limitation on growth in the open ocean. To control excessive growth of phytoplankton in the following seasons, zooplankton grazing rates were
increased. ECOSMO II has been used as an operational model for the Arctic since 2017 and its parameterization has been tested and improved various times, more than we can document here. Thus the parameter sets for EXP2 and EXP3 are provided here as milestones for ECOSMO II development. To document the development process of the ECOSMO, we present all the experiments representing DS2013 (EXP1), operational model prior to July 2021 (EXP2) and next-phase operational model (EXP3). The model evaluation is followed by an overview of the notable aspects of the simulated biogeochemistry of the North
Atlantic and the Arctic Oceans.

In addition to the 3D-HYCOM-ECOSMO II(CHL) simulations, we have also performed a 1D simulation using General Ocean Turbulence Model (GOTM; Burchard et al., 2006) using Framework for Aquatic Biogeochemical Models (FABM; Bruggeman and Bolding, 2014) as the online coupler at Station-M (66°N 2°E) in the Norwegian Sea to present the differences of ECOSMO II and ECOSMO II(CHL) both visually and statistically comparing with and without explicit chlorophyll *a* versions of ECOSMO. The details of this setup is provided in Appendix A1. Station-M is a long-term time-series station and is representative of the Norwegian Sea dynamics and data from Station-M is often used for the development of ECOSMO. The dynamics shown in Appendix A1 is representative for regions with similar plankton dynamics (e.g. Norwegian Sea, Barents Sea), thus can be used as a showcase for the new chlorophyll *a* specific addition. Apart from the improvement of model chlorophyll *a* results, the addition of dynamic chlorophyll *a* establishes a higher level of functionality of ECOSMO such that phytoplankton functional types now have their unique carbon:chlorophyll *a* ratios, initial slope of P-I curves which enables better adaptability to different environments, and the model now has better integration with observation systems (e.g. remote sensing) and future improvements toward bio-optical modelling.

## 4 Model evaluation

In this section, we present a selection of model results to provide an overview of the performance of ECOSMO II(CHL). While the model domain extends to the equatorial regions, our focus is on the Nordic Seas and the Arctic. We present the evaluation of the observable model output against in situ data with the relevant statistics. The focus of this assessment is on the key parameters of the chemical and biological fields on a regional scale where the subdomains defined for model assessment are given in Fig. 2. This approach allows for assessment of the local biogeochemical characteristics of the model. The purpose of this assessment is twofold: (1) to assess the model formulation and its parameterization as a regional hindcasting and forecasting tool, as a component of CMEMS and (2) to introduce the model as a tool for scientific studies.

The extent of the subdomains depicted in Figure 2 were defined by the geographical definitions of the regions and their environmental setting as such the BARENTS region covers the shelf area south and east of Svalbard and border NOR. N at the opening to the Norwegian Sea where it is deeper and is highly influenced by the Atlantic inflow. The Norwegian Sea is divided into north and south to take into account for the differences in daylength across the wide latitude range (20°). The border between GREENLAND and NOR. N and S. roughly locates the temperature changes of the different water masses in the region (Fig. 2). ARC regions were set to cover sea-ice covered regions most of the year. BERING STR. Region was set to separate the boundary conditions from the rest of the domain. KARA and LAPTEV regions have naturally defined borders with the islands around them. SPG region is defined to represent the subpolar gyre region.

## 4.1 Observations

The model simulations were evaluated using three different datasets as follows: (1) World Ocean Atlas 2013 (WOA13; Garcia et al., 2013), (2) Institute of Marine Research (2018) data (IMR18), (3) ESA Ocean Colour CCI v5.0 (OC CCI; Ocean Colour Climate Change Initiative; Sathyendranath et al., 2019).


The model's consistency with the large-scale climatological inorganic nutrient distributions was quantified by comparing the regionally averaged monthly inorganic nutrient model data (nitrate, silicate and phosphate) to WOA13 data . The WOA13 data were horizontally averaged in the model subdomains presented in Fig. 2. Modelled inorganic nutrients were vertically interpolated to 5 and 100 meters matching the WOA13 depth levels, spatially averaged within the subdomains and monthly
averaged in time to construct corresponding regional time-series (cf. Section 4.3; Fig. 3). These monthly time-series allowed a model evaluation for the regions, in which the in situ data was not optimal for the statistical analysis. Regional climatology data should be used with caution because WOA13 data are in some places based on very few observations and that may mislead the evaluation process. To detect the regions with low number of observations, WOA13 data points were extracted for each region and were summed up as monthly time-series (Fig. A3). As an example, the number of data points for the regions defined
as ARC (Fig. 2) were almost negligible compared to the Norwegian Sea or the Barents Sea. Further discussion on this is given in Section 4.3.

A separate evaluation for the model inorganic nutrients (nitrate, silicate and phosphate) and chlorophyll *a* was conducted using the IMR18 in situ data by performing a point-by-point (location and depth) co-location for the statistical analysis (cf. Section
4.3; Table 2). For each in situ data point, the closest model grid was selected and the vertical profile was interpolated to the observed depth. Data with only 'good' flags were used totalling to more than 120000 data points for each nutrient and chlorophyll *a*. While the size of the observed dataset is unique, the regional coverage is limited to mainly the Norwegian Sea and the Barents Sea (Fig. A5). For this reason, the analysis using WOA13 and IMR18 complement each other well with one covering wider regions and the other providing a large dataset respectively.


A final model chlorophyll evaluation was conducted using OC CCI daily surface chlorophyll *a* and downwelling attenuation coefficient at 490 nm (kd490) at 4km x 4km spatial resolution. This dataset is derived from multiple sensors: SeaWIFS, MODIS Aqua, MERIS, SeaWIFS LAC and VIIR. We used this dataset for the years 1998-2010. Chlorophyll *a* and kd490 were remapped to the model grid and the model chlorophyll *a* was averaged within 1/kd490 (m) depth. In the cases that kd490 data
were missing, the model chlorophyll *a* was averaged within 10 meters. Model chlorophyll *a* was then statistically analysed using the OC CCI chlorophyll *a*, and from this point on, OC CCI chlorophyll *a* is referred to as the satellite chlorophyll *a*. Satellite and model data covering the ocean topography shallower than 100 meters were masked out. This separate analysis

allows us to include chlorophyll *a* data for model evaluation in addition to IMR18 data. We should note that, for the North Atlantic and the Arctic, satellite data was often hindered by cloud, sea-ice coverage and winter darkness.


The analyses described above were applied to all of the experiments, EXP1, 2 and 3. Very few direct observations of primary production are available in our focus region. We have therefore used reported values from the literature for evaluating the estimated magnitude of primary production (cf. Sect. 5 for the references).

### 4.2 Statistical methods

We used the Institute of Marine Research (2018) dataset for inorganic nutrients and chlorophyll *a* to construct the statistical analyses. The statistical analyses cover 1991-2010 period and only the quality-controlled data were considered. For each in situ data point, the date and the corresponding horizontal model coordinate were identified and modelled nutrient and chlorophyll *a* were vertically interpolated to the depth of the in situ data point. We computed percent bias (% bias), root mean square error (rmse), correlation (corr) and normalized standard deviations (nstd) for the co-located data:


$$\% \, bias \, = (\textstyle\sum(M - O) * 100)/\sum O \tag{13}$$

$$rmse \, = \sqrt{\textstyle\sum(M - O)^2 / N} \tag{14}$$

$$corr \, = \left(\textstyle\sum_{i=1}^{N}(M_i - \bar{M})(O_i - \bar{O})\right)/\left(\sqrt{\textstyle\sum_{i=1}^{N}(M_i - \bar{M})^2 \sum_{i=1}^{N}(O_i - \bar{O})^2}\right) \tag{15}$$

$$nstd \, = \left(\sqrt{(\textstyle\sum_{i=1}^{N}(M_i - \bar{M})^2)/N}\right)/\left(\sqrt{\textstyle\sum_{i=1}^{N}(O_i - \bar{O})^2 / N}\right) \tag{16}$$

where M = estimated, O = observed, N = number of data points and i the individual sample. These statistics were applied to the whole simulated period but are specific to each subdomain for regional evaluations.

### 4.3 Evaluation of the model experiments

In this section, we analyse the model performance against climatology and in situ data using visual and statistical analysis. We include each experiment in the analysis as a reference for modelling studies that adopt ECOSMO II(CHL) and showcase the
possible outcomes using various parameterization sets.

Prior to presenting detailed model results, we note that the model is at a steady-state after 2-years of simulation (1989 – 1990). For the years 1991 – 2010, which we have performed our analyses, the model nitrate rate of change is 0.002, 0.0031 and -0.0007 mmolN m$^{-3}$ y$^{-1}$ for average nitrate within 0 – 100 m, 0 – 500 m, and 0 m – bottom respectively. For the same averaging

depths, the model silicate and phosphate rates of change are 0.004, 0.0055, 0.0057 mmolSi m$^{-3}$ y$^{-1}$ and 0.00004, 0.00016, -0.0026 mmolP m$^{-3}$ y$^{-1}$ respectively.

Fig. 3 depicts ECOSMO II(CHL)'s performance in representing the upper 100 m concentrations of the macronutrients nitrate, silicate and phosphate against monthly climatology. For these climatological comparisons using WOA13 data, model and observed time series are represented at the surface (5 m) and at 100 m. We note that the number of samples for the monthly climatology vary between months and regions (Fig. A3). Especially for the cases of polar regions and eastern coastal Arctic, the number of data points that were used to construct the monthly climatology were negligible compared to the remaining southern regions (Fig. 2). We have also included KARA region in the discussion here as there are significant number of data points, though limited to only late-summer (months 7 – 11). Even in the case of the Norwegian and the Barents Seas, the number of samples for winter months are significantly lower than the rest of the year.

The model is generally in good agreement with the seasonality in climatology representing the high concentrations in winter and the drawdown of nutrients in summer, but with noticeably higher winter nutrients in the Barents Sea both at the surface and at 100 m. The modelled Norwegian Sea silicate concentrations are notably higher in winter at the surface and throughout the year at 100 m. Considering the consistent agreement of modelled and observed nitrate and phosphate for the Norwegian Sea, Greenland Sea, late-summer Kara Sea and the subpolar gyre region, the simulated high silicate suggests that further tuning may be required for silicate uptake by diatoms, diatom and opal silicate sinking rates or the remineralization rates of opal. The adopted 1:1 ratio of nitrate to silicate cellular structure of phytoplankton may not be as applicable for the region. We note that although on average modelled silicate is higher than observed, occasionally diatom productivity (silicate uptake) was limited by the model formulation as silicate values approached 1 mmol m$^{-3}$ (Fig. 3). The standard deviations of both the observed and modelled nutrients are large in the case of the Barents and Norwegian Seas. The monthly modelled nutrients correspond very well with the climatological values for the surface waters in the southern regions (Norwegian Sea, Greenland Sea and SPG regions) indicating satisfactory model performance on large scale productivity and its seasonal variability in these regions. The Kara Sea is highly influenced by the coastal nutrient discharges as can be seen from the high standard deviations, especially for silicate including the late-summer where we have sufficient data for this analysis. Apart from surface silicate, the model performs generally well for the Kara Sea from month 7 and onwards. In addition to our comments about silicate above, the coastal discharge of nutrients should be improved in future studies, as in this study we used annual climatology for river nutrient discharge.

Experiments were generally comparable when the model results were regionally and monthly averaged (Fig. 3). Notable differences were found for the mid-summer nitrate and phosphate concentrations for the Barents, Norwegian and Greenland Seas, as the drawdown of these nutrients was better resolved by EXP1 compared to climatology as EXP1 summertime nutrients were lower than in EXP2 and 3. A possible reason why EXP1 has larger drawdown of nutrients during mid-summer is the

higher photosynthesis efficiency applied in EXP1 resulting in higher uptake of nutrients and higher zooplankton grazing rate applied to EXP2 and 3 resulting in higher top-down pressure to phytoplankton preventing phytoplankton from consuming more nutrients.

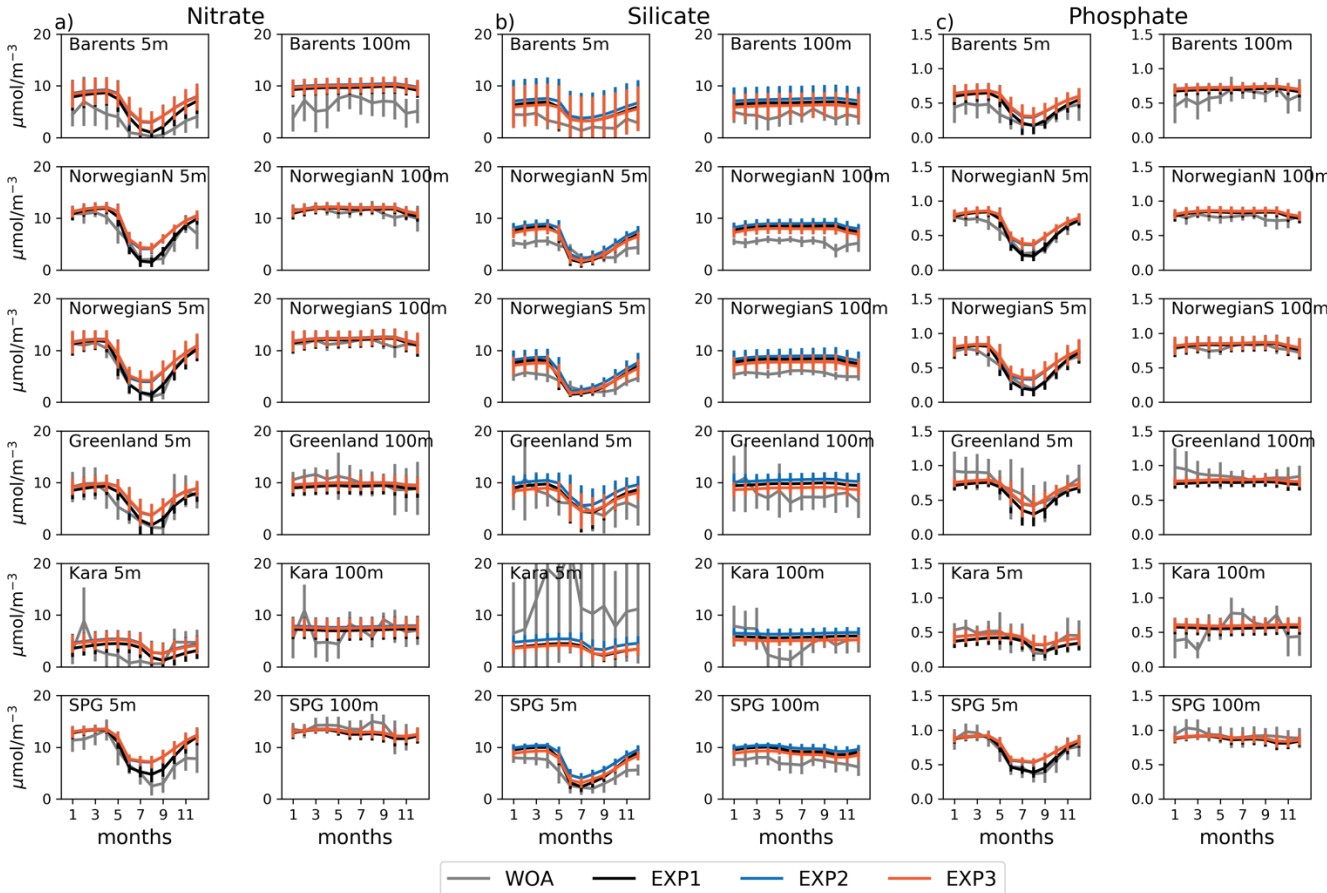

**Fig. 3: Evaluation of seasonal cycle of nutrients at 5 m and 100 m for the model (black lines) vs WOA13 (grey lines) regional monthly averages in the selected areas Barents, NorwegianS and SPG of the model domain for (a) nitrate, (b) silicate, and (c) phosphate. Model experiment (EXP1: black, EXP2: blue, EXP3: orange) and WOA13 spatial standard deviations are plotted for each month as vertical lines). The number of observations for the WOA13 time-series is given in Figures A3, A4, A5 and A6.**

With point-by-point comparison, for nitrate, model and in situ data correlations are higher than 0.8 for the three regions, with higher correlations at the higher latitudes (Table 2). One possible reason for the slight differences in correlations between lower and higher latitudes is the timing of the sampling. The majority of the sampling in the southern subdomains are held earlier in the year compared to the northern subdomains. As the model consistently initiates the spring bloom later than what is observed, a consequence of the physical model mixing scheme, it results in a later drawdown of nutrients, thus weaker correlations. The consistently occurring late spring bloom was also noted in previous Nordic Seas modelling studies using HYCOM as the physics model (Samuelsen et al., 2009 and 2015). They related the bloom-timing issue to the physics model

or the missing phytoplankton convection process of early seeding of the spring bloom by phytoplankton that was convected in
winter. However, apart from the bloom timing, correlations higher than 0.67 for silicate and 0.8 for nitrate and phosphate in general represent a good agreement on the temporal and vertical nutrient variability.

Normalized standard deviations (nstd) for nitrate are within 0.63 - 0.84 indicating that the model underrepresents the amplitude of the observed variability. The model has % biases between 0.7 – 9 % for the Norwegian Sea, whereas the bias is 13.5 – 31.3
% for the Barents Sea. For the case of root mean square error (rmse), modeled nitrate has errors between 1.94 – 3.34 mmolN m$^{-3}$. The simulated regional inorganic nutrients (EXP1) against in situ data are depicted in Fig. Fig. 4 where we make a point-by-point comparison of the modelled and observed inorganic nutrients. While the statistics include every data point, Fig. Fig. 4 depicts the upper 100 m. The observed upper 100 m nitrate maximum reach 14 mmolN m$^{-3}$ while the modelled nitrate maximum is ~11 mmolN m$^{-3}$ in the Barents Sea (Fig. Fig. 4a), whereas the nitrate maxima are similar (Fig. Fig. 4b) for the
Norwegian Sea. The source of the lower bias and rmse for the Norwegian Sea is also evident in Fig. Fig. 4b where the model to observed data points are more scattered around the 1-to-1 line compared to Fig. Fig. 4a.

**Table 2: Simulation statistics (model vs in situ) specific to each region**

| Variable | Region | CorrCoef | | | Norm. StdDev | | | Bias (%) | | | RMSE (mmol m$^{-3}$) | | |
|---|---|---|---|---|---|---|---|---|---|---|---|---|---|
| | | Exp1 | Exp2 | Exp3 | Exp1 | Exp2 | Exp3 | Exp1 | Exp2 | Exp3 | Exp1 | Exp2 | Exp3 |
| **nitrate** | Barents | 0.88 | 0.86 | 0.86 | 0.79 | 0.63 | 0.60 | 13.5 | 31.3 | 30.5 | 2.43 | 3.34 | 3.34 |
| | NorwegianN | 0.89 | 0.87 | 0.87 | 0.84 | 0.66 | 0.64 | 0.7 | 9.0 | 8.6 | 1.94 | 2.39 | 2.39 |
| | NorwegianS | 0.83 | 0.80 | 0.80 | 0.83 | 0.73 | 0.72 | 1.9 | 7.8 | 6.7 | 2.25 | 2.51 | 2.47 |
| **silicate** | Barents | 0.78 | 0.74 | 0.78 | 1.16 | 1.18 | 0.98 | 71.8 | 98.8 | 59.5 | 2.62 | 3.41 | 2.23 |
| | NorwegianN | 0.72 | 0.67 | 0.74 | 1.07 | 1.02 | 0.94 | 46.8 | 61.7 | 37.9 | 2.89 | 3.49 | 2.46 |
| | NorwegianS | 0.73 | 0.67 | 0.74 | 1.02 | 1.00 | 0.91 | 39.9 | 53.0 | 31.0 | 2.63 | 3.19 | 2.26 |
| **phosphate** | Barents | 0.90 | 0.88 | 0.88 | 0.80 | 0.63 | 0.61 | 0.7 | 13.9 | 14.7 | 0.13 | 0.17 | 0.17 |
| | NorwegianN | 0.90 | 0.88 | 0.89 | 0.94 | 0.75 | 0.72 | 0.0 | 7.1 | 7.4 | 0.11 | 0.13 | 0.13 |
| | NorwegianS | 0.83 | 0.81 | 0.81 | 0.95 | 0.84 | 0.82 | -1.4 | 3.8 | 3.8 | 0.13 | 0.14 | 0.14 |
| **chlorophyll a** | Barents | 0.38 | 0.29 | 0.3 | 0.97 | 0.61 | 0.57 | 6.2 | -62.1 | -61.7 | 0.95 | 0.92 | 0.90 |
| | NorwegianN | 0.41 | 0.33 | 0.34 | 2.06 | 1.25 | 1.22 | 68.0 | -26.7 | -25.1 | 1.49 | 1.03 | 1.00 |
| | NorwegianS | 0.23 | 0.19 | 0.20 | 1.50 | 1.06 | 0.97 | 20.3 | -36.4 | -39.0 | 1.76 | 1.46 | 1.40 |

For silicate, model and in situ data correlations (Table 2) range between 0.67 – 0.78, and, similar to nitrate, correlations are slightly higher at the higher latitudes. However, silicate variability due to uptake is only dependent on diatom productivity thus a direct relation to nitrate dynamics should not be expected. Both, the Barents and Norwegian Sea modelled maximum silicate for the upper 100 m are higher than the observations with model % biases between 31 – 98.8 and rmse between 2.23 – 3.49 mmolSi m$^{-3}$. The model performs well for the silicate nstd with values very close to 1 (0.91 – 1.18) indicating the model
represents the amplitude in silicate seasonal variability well. The model is formulated to limit the uptake of silicate with

concentrations below 1.0 mmolSi m$^{-3}$ where the effect is visible in Fig. Fig. 4c,d. The sources of high model biases and rmse's are also evident in these figures where the scattered data points are mostly below the 1-to-1 line.

The phosphate statistics are similar to those of nitrate, an expected result as all phytoplankton consume phosphate with a fixed Redfield N:P ratio. Correlations are between 0.81 – 0.9 with higher values at the higher latitudes. The nstd's (Table 2) are slightly better than those of nitrate with values between 0.61 – 0.95 indicating that the model underestimates the amplitude in phosphate variability. In agreement with the underestimated amplitude in variability, observed phosphate maximum for the upper 100m (not shown) reach 1.25 mmolP m$^{-3}$, where model maximum for all regions are ~1.0 mmolP m$^{-3}$. In terms of % biases (-1.4 – 14.7) and RMSEs (0.11 – 0.17 mmolP m$^{-3}$), the model simulates phosphate better than nitrate and silicate.


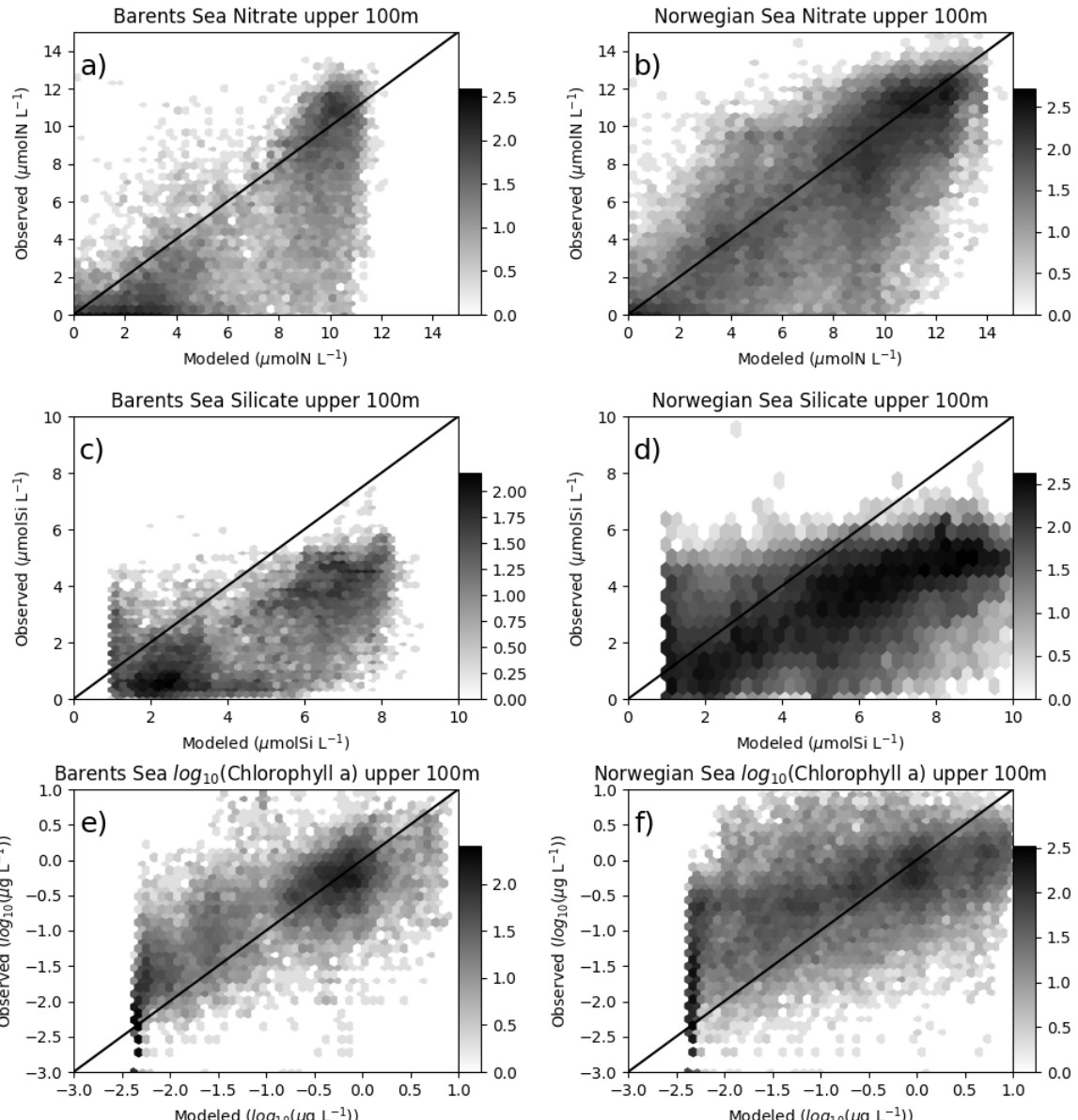

**Fig. 4: Co-located modeled (EXP1) and in situ upper 100m nitrate (a) Barents Sea, (b) Norwegian Sea, silicate (c) Barents Sea, (d) Norwegian Sea and chlorophyll (e) Barents Sea, (f) Norwegian Sea comparisons. Log10 number of points are represented in hexagonal local clusters with shades of grey. Only the upper 100 m points are plotted.**


Among the experiments, all perform very similar in terms of nutrient correlations, while for nitrate and phosphate EXP1 performs slightly better in terms of nstds, EXP3 performs slightly better for silicate for the Barents Sea and EXP2 for the Norwegian Sea, though the differences among the experiments were almost negligible. Similarly, EXP1 perform better in terms of % bias and RMSE for nitrate and phosphate, and EXP3 perform better for silicate. The slightly better performance of

EXP1 for nitrate and phosphate is also evident in summer averages when compared to climatology as mentioned before (Figs. 3a and 3c). The model performance for silicate when using monthly averages shows even less differences among the experiments, however, the EXP3 is slightly closer to WOA timeseries compared to EXP1 (Fig. 3b)

In situ chlorophyll *a* correlations for the upper 100 m (Table 2) are between 0.19 – 0.41, which are below those of inorganic nutrients. However, the model performs acceptable in terms of nstd's. For the Norwegian Sea, EXP3 has the better performance (0.97 and 1.2) and for the Barents Sea, EXP1 perform better (0.97). While EXP1 perform better for the Barents Sea and The NorwegianS (6.2 and 20.3 %) in terms of % bias, EXP3 perform better for NorwegianN (-25.1). Among all the experiments, EXP3 performed better in terms of RMSE for the three regions (0.9 – 1.4 mg m$^{-3}$). The concentration ranges (Fig. Fig. 4f) are similar (0 – 10 mg m$^{-3}$) for both the observed and modelled for the Norwegian Sea indicated by nstd's near 1.0, but the points are scattered away from the 1-to-1 line indicating the low correlations. Model chlorophyll *a* is always below 8 mg m$^{-3}$ for the Barents Sea (Fig. Fig. 4e) where the observations show values above 10 mg m$^{-3}$ indicating the lower nstd is underestimating the amplitude of variability.

## 5 Simulated biogeochemistry of the North Atlantic and the Arctic

Primary production (PP) is the foundation for all marine biological production and the most frequently observed rate in BGC models. Still, there are only few observations of primary production available in the ocean as a whole, but the high Arctic is particularly poorly sampled (Matrai et al., 2013). Because the model does not have an explicit term for respiration, we can only extract gross primary production from the model, which is then compared to observations. The modelled gross annual primary production ranges from above 200 gC m$^{-2}$ y$^{-1}$ in the southern part of the model domain to almost zero in the central Arctic and features a gradual decrease from 144.26 gC m$^{-2}$ y$^{-1}$ to 41.48 gC m$^{-2}$ y$^{-1}$ from lower latitudes (SPG) towards the higher latitudes (Barents) respectively, with a sharp decrease to very low values (<6 gC m$^{-2}$ y$^{-1}$) in the sea ice covered areas Fig. 5: Simulation averaged (EXP1) model results; (a) vertically integrated (0–200 m) annual primary production (gC m$^{-2}$ y$^{-1}$), (b) annually averaged surface chlorophyll *a* (mg m$^{-3}$), and simulation averaged vertically integrated (0–200 m) plankton functional type biomass (gC m$^{-2}$) (c) diatoms, (d) microzooplankton, (e) flagellates, (f) mesozooplankton. The colorbar to FigureTable 3: Regional vertically integrated (0–200 m) annual gross primary production (gC m$^{-2}$ y$^{-1}$) and simulation averaged vertically integrated (0–200 m) plankton functional type biomass (gC m$^{-2}$; DIA: diatoms, FLA: flagellates, MIC: microzooplankton, MES: mesozooplankton) in EXP1. See Sect. 3 for the definition of the regions. Note that the BERING STR. subdomain is within the effective area of the open boundary conditions thus is relaxed to climatology.(Fig. 5) as a consequence of light limitation. Rey (1981) estimated the primary production in the Norwegian Coastal Current to range from 90 - 120 gC m$^{-2}$ y$^{-1}$, which agrees well with the values from this model (Fig. 5Fig. 5: Simulation averaged (EXP1) model results; (a) vertically integrated (0–200 m) annual primary production (gC m$^{-2}$ y$^{-1}$), (b) annually averaged surface chlorophyll *a* (mg m$^{-3}$), and simulation averaged vertically integrated (0–200 m) plankton functional type biomass (gC m$^{-2}$) (c) diatoms,

(d) microzooplankton, (e) flagellates, (f) mesozooplankton. The colorbar to Figure), although the used coarse resolution model does not represent a very distinct coastal current. Previous studies have estimated the primary production in the Fram Strait from 50 - 80 gC m$^{-2}$ y$^{-1}$ (Hop et al., 2006), while our model show values of 90-100 gC m$^{-2}$ y$^{-1}$ in the Atlantic waters and up to 30-60 gC m$^{-2}$ y$^{-1}$ on its western side. Lee et al. (2015) compared multiple Arctic models against in situ observations. Only a few of these observations were in the central Arctic while the majority were located in the Chukchi Sea, which is very close to the zone where the model is relaxed to climatology. They found a median value of all Arctic observations of 246 mgC m$^{-2}$ d$^{-1}$ which corresponds to about 90 gC m$^{-2}$ y$^{-1}$. The regional estimates of primary production were similar, but the shelf regions were the most productive. The model results for the regions surrounding the central Arctic Ocean fall in the range of this estimate, but observation base estimates for the central Arctic, although only few are available, are higher than the model results. From Lee et al. (2015) the primary production estimates from the central Arctic varied between 10 and 100 mgC m$^{-2}$ d$^{-1}$ (~4 - 40 gC m$^{-2}$ y$^{-1}$) while the model is below 1 gC m$^{-2}$ y$^{-1}$. In the model formulation, the ice is blocking more light than what is realistic and ice leads cannot be resolved, so our estimate is expected to be low in ice covered regions. It is known that both melt ponds and leads can act as windows into the ocean, facilitating blooms (Assmy et al., 2017). The light below the ice will be improved in future versions of the model system. In situ observations in the Arctic range up to more than 5000 mgC m$^{-2}$ d$^{-1}$. The model does not reproduce the extremes in primary production, but the mean values are overall consistent with available observations.

For the Norwegian and Barents Seas, the modeled primary production show distinct seasonal patterns with almost negligible productivity between November – April due to low light availability (Fig. 6). During the onset of the spring bloom, production is notably at its highest during May – June followed by a gradual decrease towards late fall. Regional differences in primary production are also evident in year-round time-series, where the Norwegian Sea primary productivity is significantly higher than the Barents Sea productivity. The southern part of the Norwegian Sea (NorwegianS) has a notably earlier (~2 weeks) bloom compared to the northern counterpart.

Table 3: Regional vertically integrated (0–200 m) annual gross primary production (gC m$^{-2}$ y$^{-1}$) and simulation averaged vertically integrated (0–200 m) plankton functional type biomass (gC m$^{-2}$; DIA: diatoms, FLA: flagellates, MIC: microzooplankton, MES: mesozooplankton) in EXP1. See Sect. 3 for the definition of the regions. Note that the BERING STR. subdomain is within the effective area of the open boundary conditions thus is relaxed to climatology.

| Region | PP gC m$^{-2}$ y$^{-1}$ | DIA gC m$^{-2}$ | FLA gC m$^{-2}$ | MIC gC m$^{-2}$ | MES gC m$^{-2}$ |
|---|---|---|---|---|---|
| Barents | 41.48 | 0.311 | 0.243 | 0.061 | 0.847 |
| NorwegianN | 98.2 | 1.191 | 0.442 | 0.12 | 1.673 |
| NorwegianS | 89.26 | 0.82 | 0.441 | 0.108 | 1.551 |
| ArcAtl | 2.6 | 0.051 | 0.027 | 0.007 | 0.044 |
| Laptev | 2.31 | 0.063 | 0.013 | 0.004 | 0.031 |

| | | | | | |
|---|---|---|---|---|---|
| ArcEast | 0.22 | 0.024 | 0.019 | 0.002 | 0.009 |
| ArcCan | 0.17 | 0.019 | 0.016 | 0.002 | 0.002 |
| Bering STR | 6.3 | 0.117 | 0.021 | 0.01 | 0.091 |
| Kara | 5.28 | 0.094 | 0.026 | 0.009 | 0.067 |
| Greenland | 34.95 | 0.399 | 0.153 | 0.046 | 0.595 |
| SPG | 144.26 | 1.56 | 0.647 | 0.214 | 2.128 |

The simulated seasonal evolution of primary production reflects the growth of plankton functional types, with diatoms (compared to flagellates) being the dominant type in the Nordic Seas spring bloom (Fig. 7). A relatively minor flagellate bloom follows a few weeks after that of diatoms. Zooplankton biomass increase from May – June in response to phytoplankton growth and is maintained till the end of the year. Note that ECOSMO II(CHL) allows zooplankton to feed on detritus, which contribute

to zooplankton sustaining growth beyond the seasons of phytoplankton activity. Towards the lower latitudes, south of 45 °N, flagellates maintain a similar annually integrated productivity ($\sim$1.5 vs $\sim$1.0 (gC m$^{-2}$)) to that of diatoms (Fig. 5Fig. 5: Simulation averaged (EXP1) model results; (a) vertically integrated (0–200 m) annual primary production (gC m$^{-2}$ y$^{-1}$), (b) annually averaged surface chlorophyll $a$ (mg m$^{-3}$), and simulation averaged vertically integrated (0–200 m) plankton functional type biomass (gC m$^{-2}$) (c) diatoms, (d) microzooplankton, (e) flagellates, (f) mesozooplankton. The colorbar to Figurec-e).

Mesozooplankton are the dominant grazer in all regions (Fig. 5Fig. 5: Simulation averaged (EXP1) model results; (a) vertically integrated (0–200 m) annual primary production (gC m$^{-2}$ y$^{-1}$), (b) annually averaged surface chlorophyll $a$ (mg m$^{-3}$), and simulation averaged vertically integrated (0–200 m) plankton functional type biomass (gC m$^{-2}$) (c) diatoms, (d) microzooplankton, (e) flagellates, (f) mesozooplankton. The colorbar to Figured-f). Similar to primary production, the NorwegianN and NorwegianS functional type biomasses are higher compared to Barents functional type biomasses with daily

200m averaged biomasses reaching 75 – 100 mgC m$^{-3}$ for diatoms and mesozooplankton in the Norwegian Sea, and $\sim$50 mgC m$^{-3}$ for the Barents Sea respectively. For both Barents and Norwegian Sea, flagellate and microzooplankton biomasses do not exceed $\sim$25 mgC m$^{-3}$ during their highest productive seasons (Fig. 7).

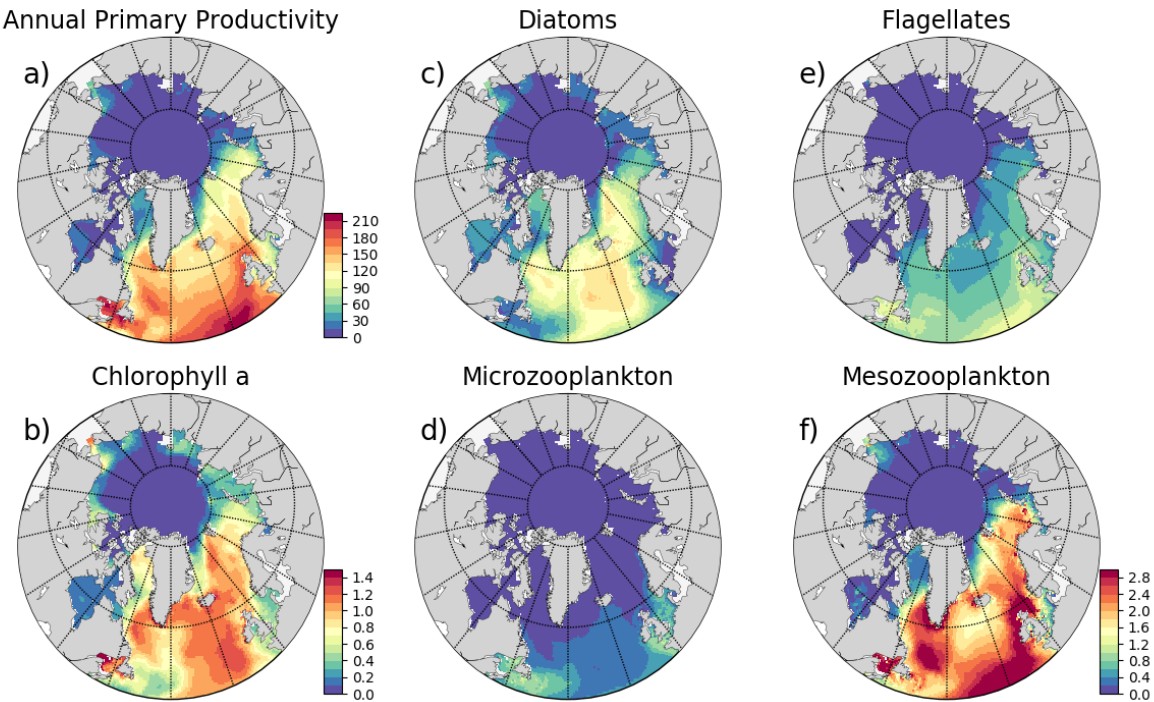

**Fig. 5:** Simulation averaged (EXP1) model results; (a) vertically integrated (0–200 m) annual primary production (gC m$^{-2}$ y$^{-1}$), (b) annually averaged surface chlorophyll *a* (mg m$^{-3}$), and simulation averaged vertically integrated (0–200 m) plankton functional type biomass (gC m$^{-2}$) (c) diatoms, (d) microzooplankton, (e) flagellates, (f) mesozooplankton. The colorbar to Figure f applies to Figures c, d, e and f.

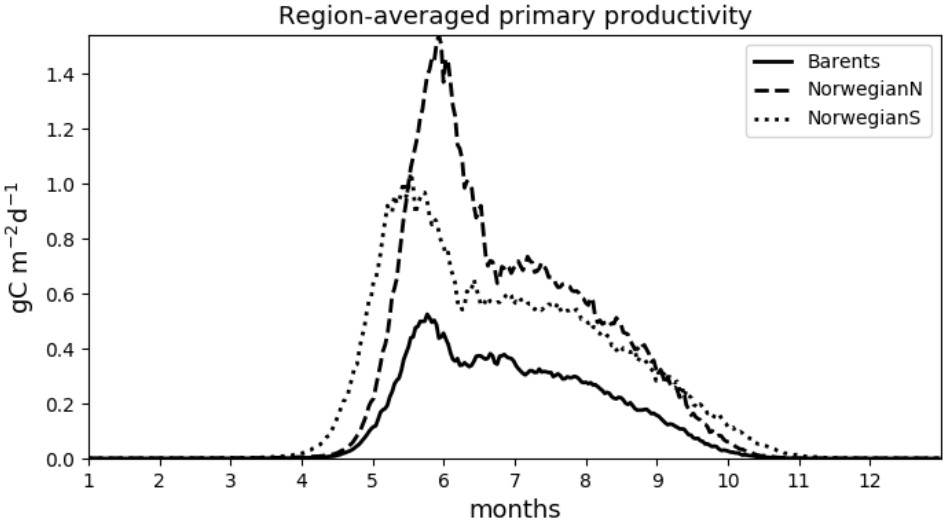

**Fig. 6:** Simulated (EXP1) time-series of 0-200 m integrated primary productivity (gC m$^{-2}$ d$^{-1}$) for different regions: (a) Barents, (b) NorwegianN and (c) NorwegianS. See Table 3 for annually averaged primary productivities.

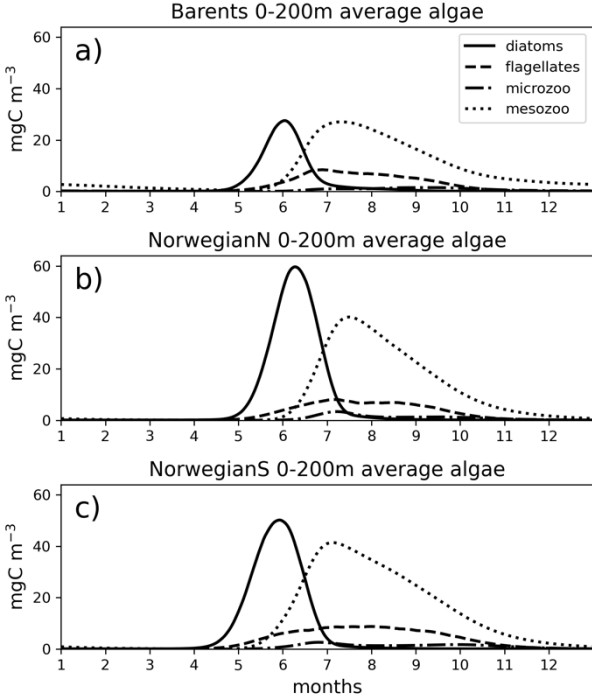

Fig. 7: Simulated (EXP1) daily averaged time-series of each plankton functional type (average of 0-200 m depth range).

The model predicts regionally high annually averaged inorganic nutrient concentrations for the subpolar gyre compared to the Norwegian and Barents Seas which is also reflected in monthly and regionally averaged concentrations (Fig. 8) with relatively lower concentrations in the coastal regions of the Nordic Seas compared to their offshore regions. The model also predicts a contrast between nutrient specific regions of high concentrations. Nitrate concentrations are higher at the lower latitudes, whereas phosphate and silicate are higher towards the higher latitudes. These features generally agree with the features of WOA2013 data (Fig. 8). As the model is relaxed towards the climatology at the Bering Strait through a sponge layer in the model domain, the overall high nutrient concentrations near the Bering Strait and especially the high silicate concentrations at the Siberian coast due to higher Si/N ratio of Pacific origin water masses compared to the Atlantic water masses, and the addition of high Si/N ratio river discharge is reflected in the modelled annual averages. As mentioned earlier, the model does not allow light to penetrate sea-ice. For this reason, the model overestimates surface inorganic nutrients compared to climatology below the sea-ice as these nutrients are not consumed by primary production, but are only affected by transport and remineralization. Overall, the model performs well in terms of N/P molar ratios ($NO_3/PO_4$; Fig. A4). Both model and climatology suggest a higher N/P ratio for the Nordic Seas and lower latitudes (~12-16). At the northern and southern Barents Sea, the climatology has a lower N/P ratio (<7) but has a high ratio at the ice-edge region (>17). In contrast, the model predicts a more regular N/P distribution with a gradual decrease from 16 to 12 from lower to higher latitudes at the Barents Sea.

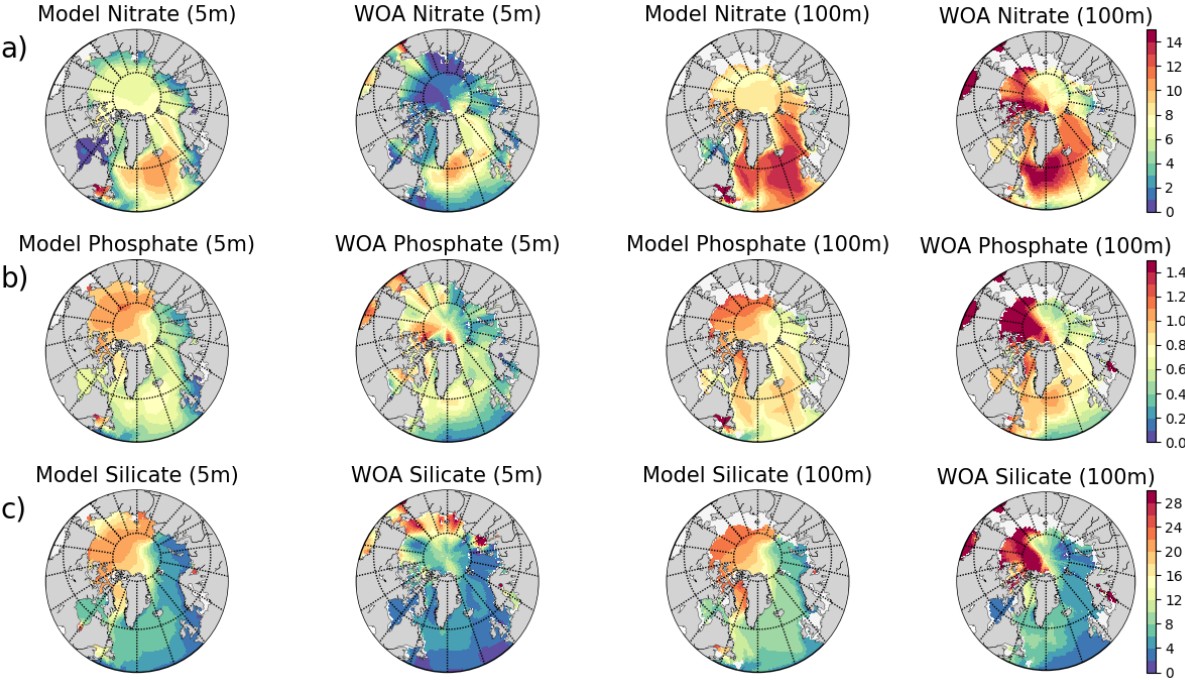

**Fig. 8: Simulation averaged (EXP1; mmol m⁻³) (a) nitrate, (b) phosphate and (c) silicate for 5 and 100 meters depth and corresponding WOA2013 annual climatologies.**

## 6 Model chlorophyll *a* against satellite data and concluding remark on experiments

Here we present the evaluation of each model experiment against satellite chlorophyll *a*. Since the parameters in EXP2 and EXP3 are used in open-ocean operational models, their performance in representing satellite chlorophyll *a* is vital for the assimilation of chlorophyll *a* in the operational model. The comparison of co-located surface in situ, model and satellite data are given in Figure 9 and their statistics are summarized in Table 4. The purpose of comparing the satellite data to both in situ

and the model is to evaluate the satellite product itself for the region, as satellite products are prone to uncertainties based on the used algorithms and are related to differences in absorption and backscattering properties of phytoplankton and concentrations of colored-dissolved organic matter (CDOM) and minerals (Dierssen, 2010). Thus, in the absence of in situ data, we have a better understanding when model and satellite data are compared. Table 5 summarizes model statistics against satellite data, which is both independent of the in situ samples and, due to the volume of satellite data, the statistics here are

based on a much more extensive dataset compared to the statistics in Table 4.

For the three regions of interest, the satellite data have a negative bias against the in situ data (Table 4). The %bias is minor for the Barents region (-5.32%), but for the Norwegian Sea, the biases are -21.34% and -16.11% for the north and south respectively. The nstd's range between 0.51 – 0.65 mg m$^{-3}$ suggesting that satellite chlorophyll $a$ underrepresents the amplitude of the in situ observed variability of chlorophyll $a$. Satellite chlorophyll $a$ rmse's range between 0.6 – 0.8 mg m$^{-3}$.

**Table 4: Estimated chlorophyll $a$ statistics against in situ surface chlorophyll $a$. Data points that are co-located with in situ data locations only are used. Co-located satellite data is also compared against in situ data for reference. See Sect. 3 for the calculation of statistics.**

| | Barents | | | NorwegianN | | | NorwegianS | | |
|---|---|---|---|---|---|---|---|---|---|
| | Bias (%) | RMSE (mg m$^{-3}$) | Norm. StdDev | Bias (%) | RMSE (mg m$^{-3}$) | Norm. StdDev | Bias (%) | RMSE (mg m$^{-3}$) | Norm. StdDev |
| Satellite | -5.32 | 0.60 | 0.65 | -21.34 | 0.66 | 0.51 | -16.11 | 0.80 | 0.53 |
| EXP1 | 39.75 | 1.21 | 1.52 | 178.25 | 2.85 | 3.50 | 140.56 | 2.74 | 2.54 |
| EXP2 | -55.69 | 1.08 | 1.15 | 50.76 | 1.87 | 2.68 | 39.59 | 1.89 | 1.85 |
| EXP3 | -51.21 | 1.03 | 1.02 | 43.32 | 1.67 | 2.43 | 23.50 | 1.66 | 1.59 |

**Table 5: Model chlorophyll $a$ statistics against satellite data. See Sect. 3 for the calculation of statistics.**

| | Barents | | | | NorwegianN | | | | NorwegianS | | | | SPG | | | |
|---|---|---|---|---|---|---|---|---|---|---|---|---|---|---|---|---|
| | Bias (%) | RMSE (mg m$^{-3}$) | Norm. StdDev | Corr Coef | Bias (%) | RMSE (mg m$^{-3}$) | Norm. StdDev | Corr Coef | Bias (%) | RMSE (mg m$^{-3}$) | Norm. StdDev | Corr Coef | Bias (%) | RMSE (mg m$^{-3}$) | Norm. StdDev | Corr Coef |
| EXP1 | 56.23 | 2.21 | 1.45 | 0.0 | 203.51 | 2.95 | 4.52 | 0.0 | 130.72 | 2.45 | 4.03 | 0.0 | 140.33 | 2.28 | 5.60 | 0.01 |
| EXP2 | -15.73 | 1.78 | 1.05 | 0.05 | 59.69 | 1.88 | 3.16 | 0.28 | 26.84 | 1.69 | 2.97 | 0.25 | 10.34 | 1.29 | 3.33 | 0.27 |
| EXP3 | -22.83 | 1.67 | 0.91 | 0.05 | 52.12 | 1.75 | 2.95 | 0.28 | 17.70 | 1.51 | 2.67 | 0.26 | 8.12 | 1.25 | 3.23 | 0.26 |

EXP1 has higher chlorophyll $a$ concentration compared to EXP2 and EXP3. This is visually evident when model and satellite data are plotted against in situ data (Fig. 9) where EXP2 and EXP3 generally form clusters distinct from EXP1. EXP1 chlorophyll $a$ are mainly located at the right side of the 1-to-1 line suggesting a positive bias against the in situ data evident in Table 4 with 39.75% for the Barents Sea and 178.25% and 140.56% for north and south Norwegian Sea respectively. The rmse's and nstd's are also higher compared to EXP2 and EXP3. Relatively, EXP1 is the least representative of the in situ chlorophyll $a$ data among the experiments. EXP2 and EXP3 overestimate chlorophyll $a$ for the Norwegian Sea with %biases ranging between 23.5% - 50.76%, and overrepresenting the amplitude of variability with nstd's ranging between 1.59 – 2.68 mg m$^{-3}$. For the Barents Sea, while EXP2 and EXP3 have negative biases and rmse's around ~1 mg m$^{-3}$, their nstd's show that they correctly estimate the amplitude of variability. With a much larger number of data points, the model error statistics computed from satellite data are similar (Table 5) with EXP1 resulting in the highest chlorophyll $a$ values statistically performing the worst compared to EXP2 and EXP3. Notably, EXP2 and EXP3 biases are much lower compared to the statistics against in situ with the exception of NorwegianN, as well as performing less errors overall. Satellite data also increase the regional coverage of the statistical analyses where SPG region statistics show that EXP2 and EXP3 outperform the EXP1 statistics (Table 5). The consistent higher bias of EXP1 compared EXP2 and EXP3 can be explained by its higher

photosynthesis efficiency (Table 1). EXP1 show a very fast primary production response to light availability during the spring bloom period with notably higher chlorophyll *a* concentrations compared to the observations (results not shown) evident in the high %biases, whereas chlorophyll *a* concentrations in EXP2 and EXP3 are closer to observed values during spring bloom. Originally, the ECOSMO II parameterization was set for the North Sea and the Baltic Sea with different light conditions. In the open ocean such a high response curve leads to an overestimate of the bloom. However, winter convective mixing is very deep in the Nordic Seas, thus the light is a limiting factor on growth. To overcome deep mixing and prevent a late spring bloom, the phytoplankton were allowed to have very high growth rates for EXP2 and relatively less higher growth parameters were set for EXP3. Statistically and visually (Fig. 9), both EXP2 and EXP3 are very similar, with EXP3 performing statistically slightly better.

The statistical analysis performed against satellite chlorophyll *a* highlights the use of satellite data as an independent dataset for model evaluation, and by its model domain-wide (though limited to surface) coverage, it allows for a more composite evaluation of the model as a whole. Satellite data is acquired in near-real time, thus presents a valuable opportunity for an operational model validation, whereas model validation with in situ data have significant delays (though very valuable for hindcast evaluation). Recent additions to satellite datasets such as the phytoplankton functional types (e.g. https://doi.org/10.48670/moi-00099) further details the use of satellite data for models. As an operational model, ECOSMO II(CHL) works well with satellite data with the inclusion of explicit chlorophyll *a* variables for each phytoplankton functional type (PFT). Not only ECOSMO II(CHL) better resolves surface chlorophyll *a* with its light-dependent dynamic carbon:chlorophyll *a* ratios (cf. Section A1), PFT specific parameters such as initial slope of P-I curves adds further details to model adaptability to varying environmental conditions (Fig. A1d) compared to an average constant ratio common to all PFTs. PFT specific model configurations further synergises with satellite PFT observations in the context of operational biogeochemical modelling. Future iterations of ECOSMO should also include such kind of evaluations. An important addition to explicit chlorophyll *a* variable is the inclusion of the initial slope of P-I curves to light-limitation on growth. In this study, light-limitation was approached in a PFT and chlorophyll *a* independent fashion (Eq. 4). Future versions of ECOSMO should adopt ways (e.g. Evans and Parslow, 1985) to include the PFT specific P-I curve slopes to take full advantage of the explicit chlorophyll a variable. This would allow PFTs to differentiate their niche light conditions for production, and further allow better integration with the bio-optical modelling of the marine environment.

We note that the statistical analysis results against satellite chlorophyll *a* contradict the statistics against in situ data (Section 4.3) as EXP1 performed better in some cases such as % bias for the Barents Sea and south Norwegian Sea. However, in the analysis against satellite chlorophyll *a*, EXP1 was statistically outperformed by EXP2 and EXP3. First possible cause of this difference may be that the in situ data and satellite data are different datasets such that they cover different locations and seasons and use different size of datapoints. In situ data were restricted in both the overall number of data points, as well as the seasons where most of the data were from late spring and onwards whereas satellite data also cover earlier parts of the year

under favourable weather conditions. As a results, model statistics may have a seasonal bias towards the timing of the in situ sampling. Second, satellite data cover only the surface of the water column where in situ chlorophyll *a* were well below the penetration depth of the satellites which might affect the statistics.

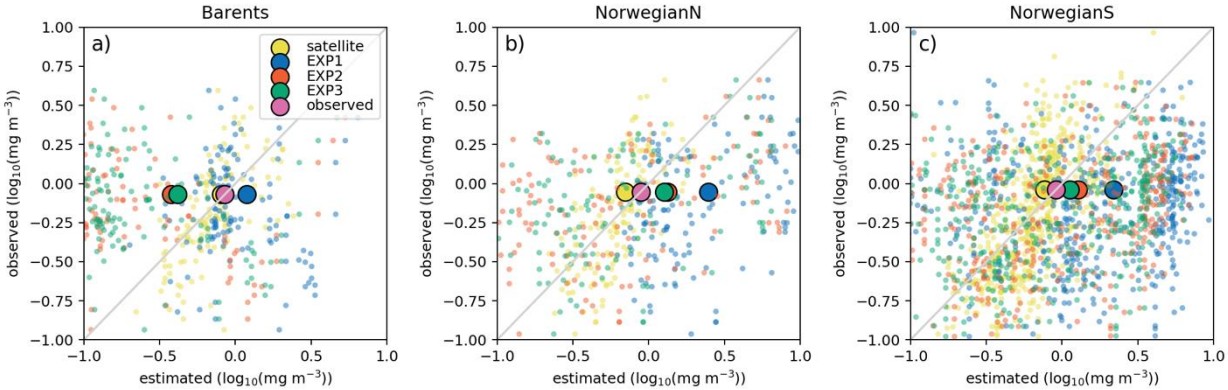

**Fig. 9: Estimated surface chlorophyll *a* data against in situ observations (log10(mg m$^{-3}$)). Region-wide averages are depicted with the large markers representative of the individual points depicted with the same colors in the background with smaller-sized markers.**

## 7 Conclusions

In this paper we present the mathematical description of ECOSMO II(CHL) which is used as the biogeochemical model for the operational forecasting of the Arctic Ocean. We document ECOSMO II(CHL) model performance with objectively analysing the model inorganic nutrients and chlorophyll *a* against available data spanning from climatology to in situ and satellite chlorophyll *a*. We compare three experiments with different parameters representing the original implementation of ECOSMO II, CMEMS Arctic operational ECOSMO II(CHL) for the years 2016 – 2021, and the current (since June 2021)

operational ECOSMO II(CHL). Through presenting the model description and its evaluation, we document the performance of ECOSMO for each of these use cases for the users of the model. While each setup perform better for some variables or datasets, the qualitative and quantitative evaluation of the model results of inorganic nutrients, chlorophyll *a* and primary production for each case demonstrated that the model is consistent with the large-scale climatological nutrient variability, and is capable of representing regional and seasonal changes. The model primary production agrees with previous measurements.

ECOSMO II(CHL) benefits from the use of explicit definition of phytoplankton functional type chlorophyll *a* implementation, i.e. the use of phytoplankton-specific dynamic chlorophyll *a*-to-carbon ratios in reference to a fixed ratio in the original model, with improved surface estimations of chlorophyll *a*, and gains added value towards improving model evaluation opportunities using satellite observations and phytoplankton functional type specific additions to model structure. In its current state, ECOSMO II(CHL) with its intermediate complexity definition of the North Atlantic and Arctic Ocean ecosystem structure

including a sediment layer is a capable modelling tool for both scientific and operational use. The modelling structure presented in this study, ECOSMO II(CHL), including the physical model, HYCOM, forms the basis of the modelling framework that the future updates will build on.

**Appendices**

**A.1 Comparison of ECOSMO II and ECOSMO II(CHL) chlorophyll *a* dynamics at Station-M**


In this section we present a 1D model setup at Station-M (66ºN 2ºE) in the Norwegian Sea using GOTM as the physics model using 1-hour interval atmospheric forcing. The location of the station resides in the Norwegian Sea South region depicted in Figure 2. We performed a 27-year run starting in 1990 using WOA2013 profiles from January climatology for the biogeochemical variables and considered the first 5 years as the spin-up period. Model results and statistics provided in Figure

A1 and Table A1 are calculated from the last 22 years. Statistical analysis was performed using the Station-M time-series data which is included in the Institute of Marine Research (2018) dataset described in 4.2. We assumed a carbon:chlorophyll *a* ratio of 60 for ECOSMO II to perform the analyses using the total phytoplankton biomass. The chlorophyll *a* depiction from ECOSMO II therefore indicate only the phytoplankton biomass and does not affect the model in any way, whereas in the case of ECOSMO II(CHL), chlorophyll *a* is explicitly represented for each phytoplankton type and the results are real model

chlorophyll *a* outputs. EXP3 parameters are used for these simulations.

The major difference between the 2 variants of ECOSMO II is that in the case of CHL variant, the model carbon:chlorophyll *a* ratio adapts to the light availability, where abundant light results in a higher ratio (days 140 – 250; Fig. A1d) at the surface, lower ratio in case of lower light availability either due to seasons or high attenuation due to high chlorophyll *a* at the surface.

The latter case can be observed around day 150 (Fig. A1d).

A significant difference in the results is that the non-CHL variant simulates higher chlorophyll *a* concentrations (Figs A1a,b, and c) assuming a 60 carbon:chlorophyll *a* ratio which is a representative average ratio for most of the productive period for ECOSMO II(CHL) (Fig. A1d). The difference is more pronounced in the upper 10 meters due to higher carbon:chlorophyll *a*

ratio (~100) under abundant light. While both simulations are statistically similar in general, especially in the deeper euphotic zone (40 – 80 m), ECOSMO II(CHL) statistically performs better at 0 - 20 m and 20-40 m range (Table A1) for almost all statistical quantities. The data that was used to calculate the statistics in Table A1 is visualised in Figure A2 using a scatter plot of the modelled and observed chlorophyll *a*, which confirms the values in Table A1 showing a slightly better performance of ECOSMO II(CHL) near the surface (0 - 20 m range). 30 m average point is visually slightly better for ECOSMO II, which

probably reflects the better % bias performed for 20 – 40 m range (Table A1). While overall the model performance improves,

further modifications to either model parameters or formulation should be done for the future iterations of ECOSMO as below 40 meters, the model has not gained a significant improvement suggesting that the chlorophyll *a* dynamics should be improved for low-light conditions.

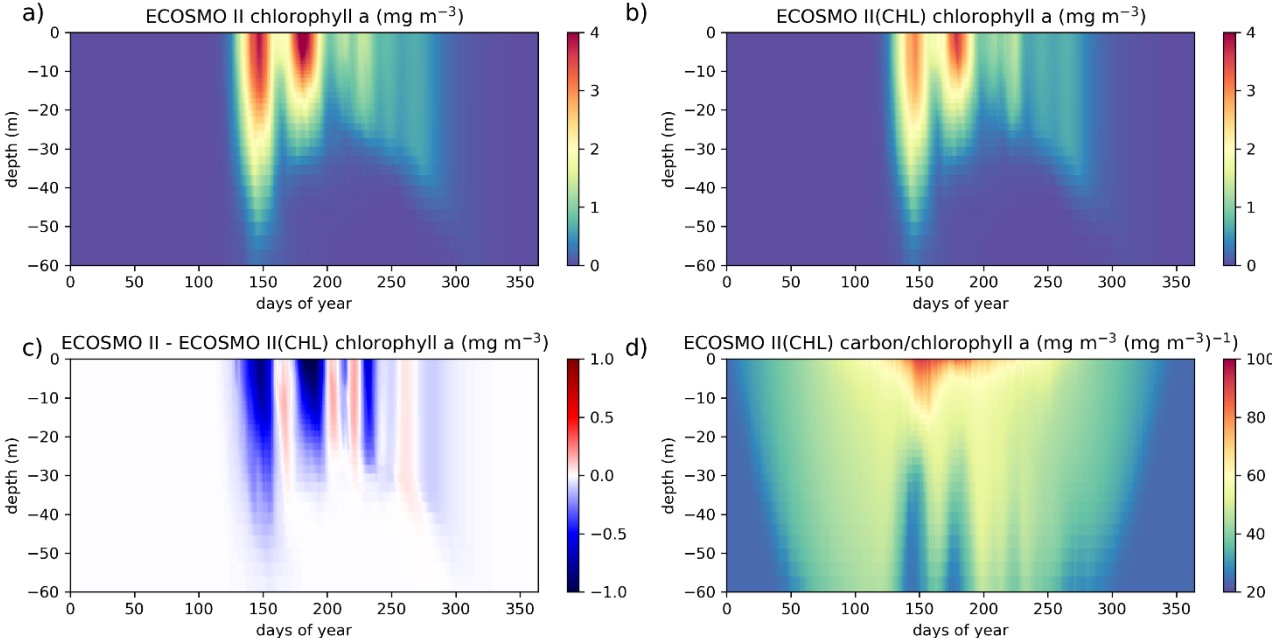

**Fig. A1: ECOSMO II chlorophyll *a* seasonal evolution is compared to ECOSMO II(CHL) using a 27-year (1990 – 2016) 1D simulation at Station-M (66ºN 2ºE) in the Norwegian Sea. Results provided here are the averages of the last 22 years (1995 – 2016) of the simulations given as annual climatologies. Figures a and b depict chlorophyll *a* concentrations of ECOSMO II and ECOSMO II(CHL) respectively, c depicts the chlorophyll *a* difference of the 2 simulations, and d depict the diatom and flagellate averages of carbon:chlorophyll *a* ratios.**


**Table A1: Comparison of ECOSMO II and ECOSMO II(CHL) chlorophyll *a* statistics against in situ data depicting 20 m sections of the upper 80 m water column using an output from a 1D model simulated at Station-M (66ºN 2ºE) in the Norwegian Sea.**

|  | ECOSMO II | | | | ECOSMO II(CHL) | | | |
|---|---|---|---|---|---|---|---|---|
|  | Bias (%) | RMSE (mg m⁻³) | Norm. StdDev | CorrCoef | Bias (%) | RMSE (mg m⁻³) | Norm. StdDev | CorrCoef |
| 0 – 20 m | 42.76 | 1.27 | 2.14 | 0.38 | 21.97 | 1.05 | 1.78 | 0.39 |
| 20 – 40 m | -23.96 | 0.68 | 1.25 | 0.40 | -33.37 | 0.62 | 1.06 | 0.42 |
| 40 – 60 m | -62.99 | 0.49 | 0.72 | 0.26 | -68.35 | 0.48 | 0.61 | 0.28 |
| 60 – 80 m | -89.75 | 0.18 | 0.15 | 0.44 | -91.73 | 0.19 | 0.12 | 0.45 |

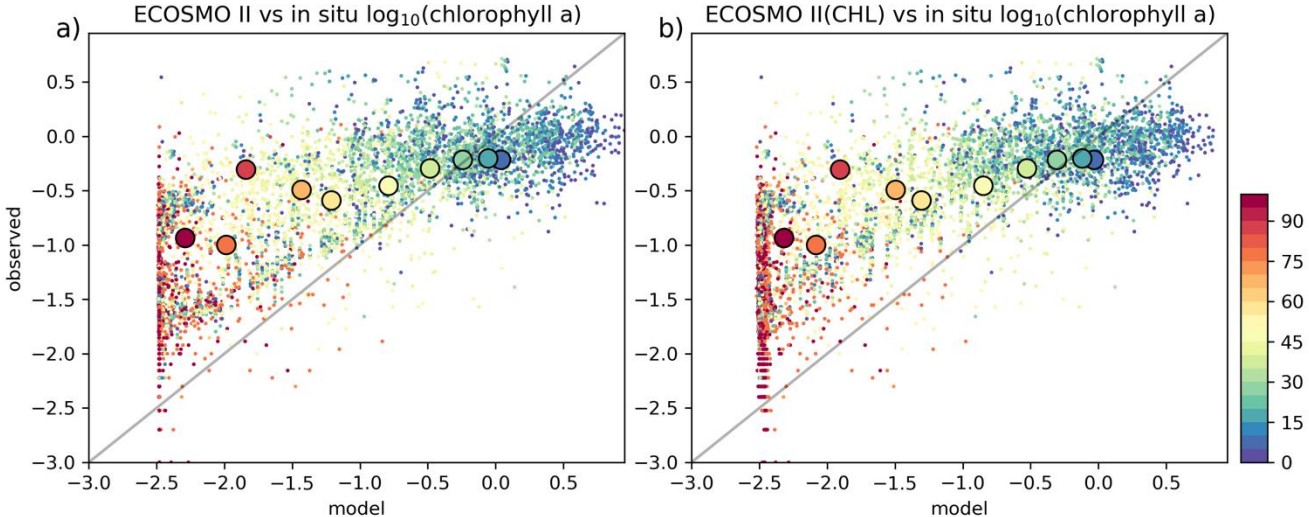

**Fig. A2: (a) ECOSMO II and (b) ECOSMO II(CHL) 1D model chlorophyll *a* (log10(mg m⁻³)) is evaluated against in situ data at Station-M. Model data was interpolated to co-locate with the in situ data. Small markers depict the individual points and the large markers depict 10 meter interval averages. The observation depth is given in color coding.**

**A.2 World Ocean Atlas 2013 and Institute of Marine Research (2018) data supplementary figures**

In this section we provide the supplementary figures for Section 4.3 and 5 by presenting the number of observations used for the statistical analyses in WOA2013 dataset for each region and inorganic nutrient (Fig. A3) and annual averages of $NO_3/PO_4$ molar ratios (Fig. A4), and profile locations for the IMR18 dataset (Fig. A5).

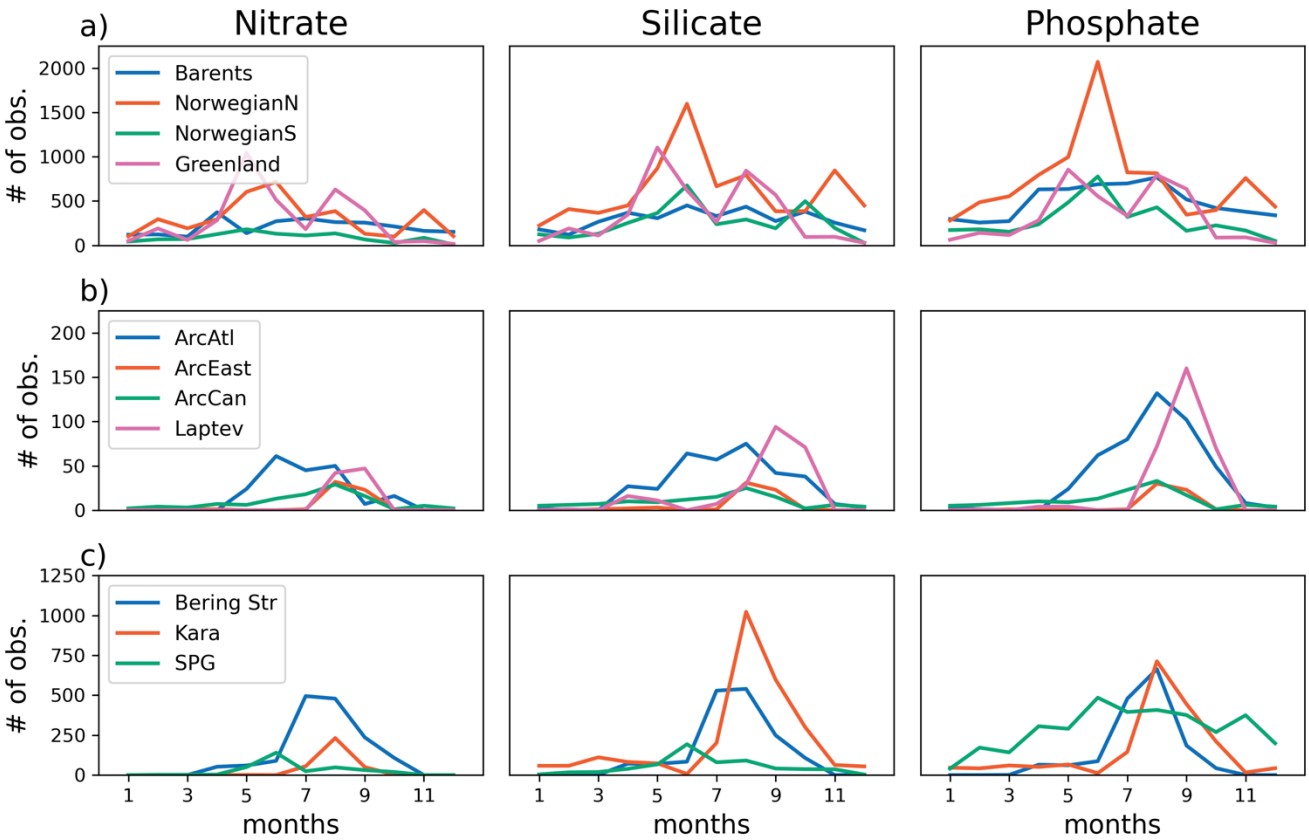

**Fig. A3: Number of observations for the WOA13 inorganic nutrient time-series for the model regions.**


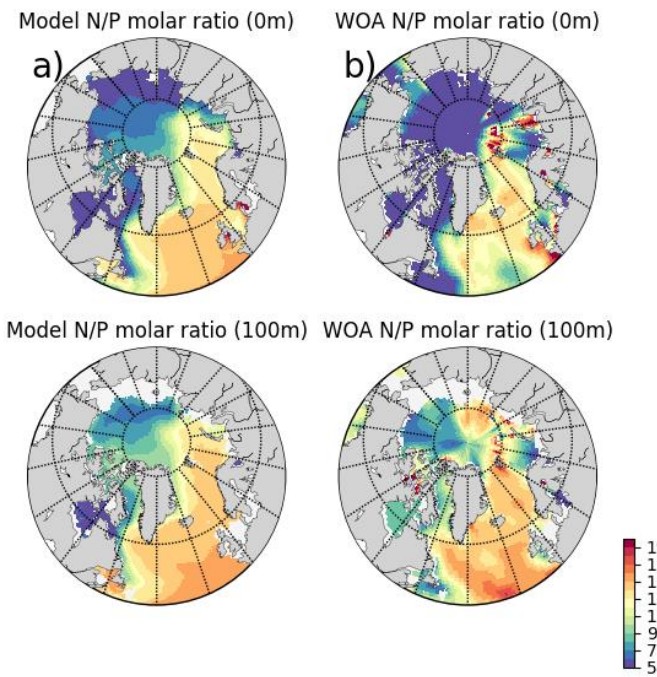

**Fig. A4: Simulated and WOA2013 inorganic nutrients annual averages NO₃/PO₄ molar ratios, a) model, b) WOA2013 for 5 and 100 meters isodepth.**

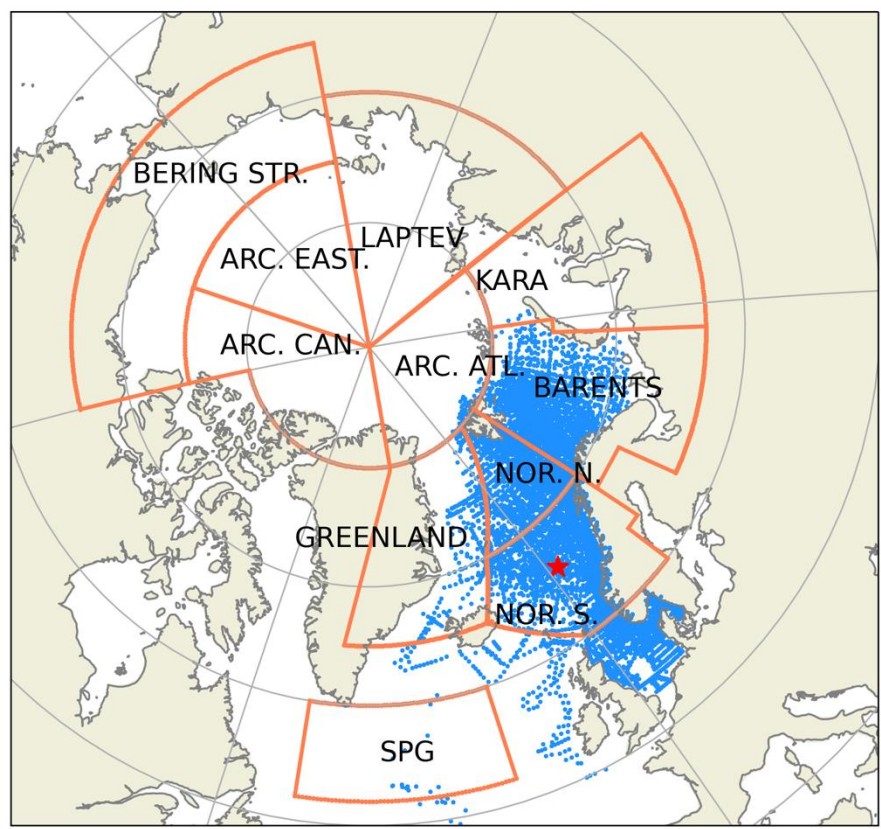

Fig. A5: Subdivision of model domain in prescribed geographical subdomains used for model quality assessments. The subdomains are as follows: Norwegian Sea South (NOR. S.), Norwegian Sea North (NOR. N.), Barents Sea (BARENTS), Kara Sea (KARA), Laptev Sea (LAPTEV), Bering Strait (BERING STR.), Arctic-Canada (ARC. CAN.), Arctic-East (ARC. EAST), Arctic-Atlantic (ARC. ATL.), Greenland Sea (GREENLAND) and the Subpolar Gyre (SPG). The points in the oceanic regions denote the profile locations for the observed biogeochemical variables that were used for the statistical analyses. The star depicts the coordinates of the Station-M time-series location. While the model domain extends down to the equatorial regions, the figure focuses on the area of interest. Note that the BERING STR. subdomain is within the effective area of the open boundary conditions thus is relaxed to climatology.

## A.3 Observational data sources

The following web links are to sources of the observational data used in the evaluation of ECOSMO II(CHL):

(1) World Ocean Atlas 2013: nitrate, phosphate and silicate

https://www.nodc.noaa.gov/OC5/woa13/ (last access: 25 March 2022)

(2) World Ocean Atlas 2018: temperature

https://www.nodc.noaa.gov/OC5/SELECT/woaselect/woaselect.html (last access: 25 March 2022)

(3) Institute of Marine Research (2018) data: nitrate, silicate, phosphate and chlorophyll *a*

http://www.imr.no/forskning/forskningsdata/infrastruktur/viewdataset.html?dataset_id=104 (last access: 25 March 2022)

(4) Ocean Colour Climate Change Initiative v5.0: chlorophyll *a*

https://climate.esa.int/en/projects/ocean-colour/data/ (last access: 25 March 2022)


## Code availability

The exact version of the model used to produce the results used in this paper is archived on Zenodo (https://doi.org/10.5281/zenodo.6387608; Lisæter et al. (2021)) including the input data and scripts to run the model and produce plots for all the simulations presented in this paper. They are openly available under Creative Commons Attribution

4.0 International license. HYCOM version used is 2.2.37 and the ECOSMO II(CHL) code is available in HYCOM_2.2.37/CodeOnly/src_2.2.37/nersc/ECOSMO where m_ECOSM_biochm.F is the master biogeochemical code. The model setup used here is located under "model_experiment/expt_09.0/SCRATCH/" directory. After the compilation following the procedure documented in "Doc" folder, the executable copied to the SCRATCH folder should be able to replicate the model presented here. The different parameters given for each experiment in the manuscript can be applied to

"HYCOM_2.2.37/CodeOnly/src_2.2.37/nersc/ECOSMO/ECOSMparam1.h". The model is set to produce daily averaged binary files, but scripts to convert the binary files to netcdf files are included in "MSCPROGS/src". The model code is written in FORTRAN. Model results provided in the manuscript are located under "model_output" directory.


## Author contribution

VCY and AS designed the experiments and VCY carried them out. UD is the developer of ECOSMO II and together with AS,

they coupled ECOSMO II to HYCOM physics model. VCY has built the ECOSMO II(CHL) version on ECOSMO II. The physics model setup for this study was mainly prepared by AS, and the preparation of the biogeochemical setup, sensitivity analyses and model evaluation were carried out by VCY. VCY prepared the manuscript with contributions from all co-authors.

## Competing interests

The authors declare that they have no conflict of interest.

**Acknowledgements**

VCY and AS acknowledge the support of CMEMS for the Arctic MFC. UD was supported through ZOOMBI (CMEMS 66-SE-CALL2). The computations were performed on the Norwegian Sigma2 infrastructure under the projects NN9481K and NS9481K. Visualization of model results and revision of this manuscript is also supported by the European Space Agency through the Cryosphere Virtual Laboratory (CVL, grant no. 4000128808/19/I-NS)

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
