# Peer review of "ECOSMO II(CHL): a marine biogeochemical model for the North Atlantic and the Arctic"

_Geoscientific Model Development, 2021_

## Author Comment (AC1)

We thank the reviewers for their positive comments. In the following, we report our proposed replies to the specific points from each referee. If accepted, these changes will be properly included in the revised version.

Response to Reviewer #1:

*The paper presents and evaluates an updated version of the marine biogeochemical model ECOSMO II, that now also includes a parameterisation for a variable C:Chl ratio of phytoplankton. The biogeochemical is coupled to the HYCOM ocean model, configured for the Arctic Ocean. Model evaluation of a default setup is carried out against observed nutrients and Chl a from in situ measurements and remote sensing, and complemented by comparison to literature values of primary production. Model sensitivity to biogeochemical parameters is evaluated in two further experiments. In general, all model configurations perform quite well with regard to the observations, and improvement in surface chlorophyll when parameters are adapted to the hydrodynamics of the Arctic.*

*In general, the paper is well written and I very much appreciate the thorough comparison to observed tracers. One drawback is the - to my eyes - somewhat incomplete description of phytoplankton (see (1) below), and the slightly confusing description of how observations were used for comparison (2). I further encourage the authors to improve on the structure of the paper (3) and to extend a bit on the comparison of the three different experiments (see my comment no 4 below).*

*(1) For better understanding of how phytoplankton responds to nutrient and light limitation I suggest to present the equation(s) for phi (in Eqns 1 and 2). Without knowledge about this functional form it is difficult for the reader to understand its dynamics, in particular as DS2013 might not be accessible to everyone. Further, does the C:Chl ratio (or the amount of chlorophyll in phytoplankton) affect the light attenuation and P-I curve, i.e. the of phytoplankton growth rate?*

We agree that the term 'phi' (growth limitation) is necessary to be present in a model description paper and understand the referee's concern that the DS2013 paper may not be accessible for everyone. For this reason, we will include $\phi_{P_j} = \min\left(\alpha(I), \beta_N, \beta_P, \beta_{Si}\right)$ as the general equation for phi and expand this equation to include all of the necessary formulations for each individual term in the equation totalling to additional 6 equations. We believe these additions will address the request of the referee. The additional equations are as follows and each equation define a limitation respective to light or nutrient:

$$\alpha(I) = \tanh\left(\varphi I(x, y, z, t)\right)$$
$$\beta_N = \beta_{NH_4} + \beta_{NO_3}$$
$$\beta_{NH_4} = NH_4/(NH_4 + r_{NH_4})$$
$$\beta_{NO_3} = (NO_3/(NO_3 + r_{NO_3}))\exp\left(-\gamma NH_4\right)$$
$$\beta_{PO_4} = PO_4/(PO_4 + r_{PO_4})$$
$$\beta_{Si} = Si/(Si + r_{Si})$$

This following equation will replace the PAR equation in the manuscript following '*In relation to the addition of a prognostic chlorophyll a state variable, photosynthetically active radiation I(x,y,z,t) at depth undergoing attenuation was modified to have chlorophyll a in the exponential term:*'

$$I(x, y, z, t) = 0.42 * I_s(x, y) exp \left( -k_w z - k_{Chl} \int_z^0 \sum_{j=1}^2 Chl_{P_j} \partial z \right)$$

For the reader to follow better, PAR will be removed from the equations and will be given as $I(x, y, z, t)$ suggesting a point in 3D and time. PAR is dependent on surface radiation, $I_s$, water and chlorophyll specific attenuation constants and CHL concentration. C:Chl ratio indirectly affects the light attenuation as Chl is included there. We believe the equation in this form better relates the reader with CHL vs light attenuation.

The model does not relate CHL to P-I curve at the moment, but it will be included in future iterations of ECOSMO.

*(2) The description of type of observations, and how these were used for model evaluation (section 4.1) is somewhat confusing (see also below, specific comments).*

Also

*Line 220-221: "Nitrate, silicate, phosphate and chlorophyll a in situ data from Institute of Marine Research (2018) were used for the statistical evaluation of the model results." Where were these data used and what is the difference to the comparison against WOA2013 data?*

Reviewer-3 has also raised this issue. We will explain the use of validation data better in Section 4.1 opening the section with the following:

'*the model simulations were evaluated using three different datasets, each of which comprises observed inorganic nutrients and chlorophyll a : (1) World Ocean Atlas 2013 (WOA13; Garcia et al., 2013), (2) Institute of Marine Research (2018) data (IMR18), (3) ESA Ocean Colour CCI v5.0 (OC CCI; Ocean Colour Climate Change Initiative; Sathyendranath et al., 2019).*'

This will inform the reader from the start what data is used. In the following paragraphs, we will go through each of them in detail and explain how and for what purpose they were used. In summary, WOA13 is used for evaluating monthly and regional averages, whereas the IMR dataset is used to evaluate the model point-by-point collocating model against in situ data points. The extensive data from IMR18 allowed us to perform the statistical analysis whereas WOA13 was used for a visual comparison. OC CCCI data was used for an additional statistical analysis supporting IMR18 data. To further assist the structure of the observed data, we will add the following figure in the Appendix which includes the profile locations for the IMR data.

[Figure]

We would like to answer item 3 and 4 and the following comments together.

*(3) Some sentences of the model setup (section 3) already anticipate the results (e.g., lines 183-186: "Using lower ... column stabilizes." and 189-191: During the continuous ... will replace the parameterization set in EXP2."). I would suggest to reorganise the paper a bit (see also comment (4)), and draw conclusions only from the model evaluation.*

Sentences anticipating the results will be removed. Please also refer to our extensive comments on the subject below.

*(4) I enjoyed reading the analysis in section 4.3. I think this analysis could be complemented by constrasting it with the results obtained with EXP2 and EXP3. For example, Table 2 shows that EXP1 always performs best with regard to the correlation coefficient, in 7 (8) cases out of 12 with regard to the normalised StdDev (Bias%), and in half of the cases with regard to RMSE. Likewise, it performs always best (regardless of metric) for phosphate and nitrate, and in half of the cases with regard to chlorophyll. Even if some differences to EXP2 and EXP3 are only small, I think this could be discussed a bit more (so far, there seems only be a sentence in line 451), and contrasted with the outcome presented in Tables 4 and 5, which indicate that EXP3 performs best with regard to surface Chl a. What is the reason for these differences? Is it because Table 4 and 5 only refer to surface values?*

*Line 278: "0.6-0.72" - Table 2 gives 0.79-0.83 for EXP1: confusion with  EXP3?*

*Line 279ff: For the biases Table 2 gives the relative bias, but in the text you refer to absolute biases, which is somewhat confusing.*

*Line 280: "2.47-3.34" - this seems to be the bias of EXP3.*

*Line 290: "0.74-0.78" - again, for EXP3?*

*Line 300: "0.81-0.89" - again EXP3?*

*Line 311: "0.97-1.2"  - EXP3?*

*Line 451: "EXPeriment statistics for inorganic nutrients are very similar in all experiments (Table 2)." - Overall, to me it seems as if EXP1 performs better than EXP2 and EXP3 (see above my comment no. (4))*

Referee #2 has also raised similar concerns. We agree that our review of the model results and discussions on the statistics of different experiments were misleading. Therefore, the text on Section 4.3 will be extensively reviewed. Going through the comments and the manuscript, we understand how the text reads to be in favor of EXP3 which was not our intent. We value all the experiments as the parameter sets in these experiments are being actively used, and our intention in this manuscript was to document the model performance for each experiment. For this reason, along with many other revisions in the manuscript, we will be clearly stating the use cases for each experiment in the introduction and the model setup, we will reference what each experiment represents (i.e. EXP1 for the original ECOSMO, EXP2 for the operational model prior to July 2021 and EXP3 for the current operational model after July 2021).

To achieve an objective comparison among the experiments, the text regarding the statistics to which the referee refers now will include results from all the experiments, not only EXP1 which will also correct the confusion stemming from EXP1 vs EXP3 as stated by the referee. We have also noted the EXP1 better performs well in many aspects and each of these will be documented in the text for both the statistical analysis and the monthly average comparison to WOA (as we will also note in the text that EXP1 averages perform better for nitrate and phosphate) and discuss possible reasons. EXP2 will also be included in these new additions.

The contrast between performances among EXP1 and EXP3 with reference to statistical analysis against IMR and satellite data will be discussed. Here are some examples we plan to add:

*"Experiments were generally comparable when the model results were regionally and monthly averaged. Notable differences were found for the mid-summer nitrate and phosphate concentrations for the Barents, Norwegian and Greenland Seas, as the drawdown of these nutrients was better resolved by EXP1 compared to climatology as EXP1 summer nutrients were lower than in EXP2 and 3. A possible reason why EXP1 has larger drawdown of nutrients during mid-summer is the higher photosynthesis efficiency applied in EXP1 resulting in higher uptake of nutrients and higher zooplankton grazing rate applied to EXP2 and 3 resulting in higher top-down pressure to phytoplankton preventing phytoplankton biomass to uptake more nutrients."*

*"Among the experiments, all perform very similar in terms of nutrient correlations while for nitrate and phosphate, EXP1 performs slightly better in terms of stds, EXP3 performs slightly better for silicate for the Barents Sea and EXP2 for the Norwegian Sea, though the differences among the experiments were almost negligible. Similarly, EXP1 perform better in terms of %*

*bias and RMSE for nitrate and phosphate, and EXP3 perform better for silicate. The slightly better performance of EXP1 for nitrate and phosphate is also evident in summer averages when compared to climatology as mentioned before (Figs. 3a and 3c). The model performance for silicate when using monthly averages shows even less differences among the experiments, however, the EXP3 is slightly closer to WOA timeseries compared to EXP1 (Fig. 3b)."*

*Line 36: Here I would prefer a more specific link to a model application at copernicus (I had to search around a bit).* The DOI will be added.

*Line 172: Evaluation after just two years of spinup seems to be quite soon - do you have any indication if the model drift (with regard to the BGC components) has decreased to some specific value?*

Below we provide model average (spatial and vertical) evolution of nutrients between 1991 and end of 2010 during which the statistical analyses were performed. Inspecting the first years, the model does not have any abnormal drifts that could be interpreted as a the spinup prior to the analyses were sufficient. We note that, unless the referee has objections, we will not include the following figures in the manuscript.

[Figure]

Line 201-202: *"The dynamics shown in Appendix A1 is expected to be valid for each model point."* - How is this expectation justified?

The sentence will be changed with the following: *"The dynamics shown in Appendix A1 is representative for regions with similar plankton dynamics (e.g. Norwegian Sea, Barents Sea), thus can be used as a showcase for the new chlorophyll a specific addition."*

Line 229 *"Processed model chlorophyll"* - What does *"processed"* mean in this context? (we were aiming for the meaning 'post-processed', but the sentence is modified to: "We used this

dataset for the years 1998-2010. Chlorophyll a and kd490 were remapped to the model grid and the model chlorophyll a was averaged to 1/kd490 (m) depth.")

*Figure 3 caption and elsewhere: "WOA18" - above you refer to WOA2013* (The text is made consistent throughout and states WOA13).

*Line 383-383: "silicate concentrations ... in the climatological data." - are discharge rates into the Arctic very different for different types of nutrients? (I.e., outside the assumed stoichometry)? Perhaps a note on this would be interesting ...*

We will add: *"As the model is relaxed towards the climatology at the Bering Strait through a sponge layer in the model domain, the high overall nutrient concentrations near the Bering Sea and especially the high silicate concentrations at the Siberian coast due to higher Si/N ratio of Pacific origin water masses compared to the Atlantic water masses, and the addition of high Si/N ratio river discharge is reflected in the model annual averages."*

*Lines 443-446: It is not really clear to me what you want to say with this sentence - could it be rephrased?* (will be rephrased to: EXP1 show a very fast primary production response to light availability during the spring bloom period with notably higher chlorophyll a concentrations compared to the observations (results not shown) evident in the high %biases, whereas chlorophyll a concentrations in EXP2 and EXP3 are closer to observed values during spring bloom.)

*Lines 478-479: "using realistic ... spin-up period" - what are the realistic constant values? and: constant with respect to what - over time? over depth?* (we agree that the sentence is misleading. We will correct it with: "We performed a 27-year run starting in 1990 using WOA2013 profiles from January climatology for the biogeochemical variables ... ")

Other minor comments will be corrected following the referee's suggestions.

Response to Reviewer #2:

*The paper presents a description and evaluation of a new configuration of the biogeochemical model ECOSMO II (ECOSMO II(CHL)) in which they have included chl in the three functional phytoplankton types as prognostic variables. Furthermore, the paper presents a comparison of model experiments using three different parameter sets. In the appendix, the authors have also included a comparison between ECOSMO II and ECOSMO II(CHL) in a configuration where the models are coupled to the 1d physical model GOTM.*

*The paper is well written and the methodology ambitious. However, I would like to have seen a larger focus on the comparison between ECOSMO II and ECOSMO II(CHL). Furthermore, apart from two sentences (451-453) I find no discussion on why EXP1 seems to perform better*

*for nitrate and phosphate (Fig 3 and Table 2) than EXP 2 and 3. I find that the paper could be published in GMD after some minor revisions.*

*Lines 139-144. What limited the concentrations from becoming too small before you added this? Also, you state that: "The minimum concentration at which the loss terms are calculated are…". Should it say: " The minimum concentration at which the loss terms are switched off"?*

There was no limit before the addition of the on/off switch, which quickly became a problem for a well-mixed open ocean. Hence we added this modification. It was a necessity for preventing a very late spring bloom, as the biomasses reached almost 0 after every winter. We agree with your suggestion and will rephrase the sentence accordingly.

*Fig.3: It is not obvious to me that EXP2 and 3 perform better than EXP1. It looks the opposite from this figure.*

Referee #1 has also raised similar concerns. We would like to share the same answer with both Reviewer #1 and #2. We agree that our review of the model results and discussions on the statistics of different experiments were misleading. Therefore the text on Section 4.3 will be extensively reviewed. Going through the comments and the manuscript, we understand how the text reads to be in favor of EXP3 which was not our intent. We value all the experiments as the parameter sets in these experiments are being actively used, and our intention in this manuscript was to document the model performance for each experiment. For this reason, along with many other revisions in the manuscript, we will be clearly stating the use cases for each experiment in the introduction and the model setup, we will reference what each experiment represents (i.e. EXP1 for the original ECOSMO, EXP2 for the operational model prior to July 2021 and EXP3 for the current operational model after July 2021).

To achieve an objective comparison among the experiments, the text regarding the statistics that the referee refers to now will include results from all of the experiments, not only EXP1 which will also correct the confusion stemming from EXP1 vs EXP3 as stated by the referee. We have also noted that EXP1 performs better in many aspects and each of these will be documented in the text for both the statistical analysis and the monthly average comparison to WOA (as we will also note in the text that EXP1 averages perform better for nitrate and silicate) and discuss possible reasons. EXP2 will also be included in these new additions.

The contrast between the performances of EXP1 and EXP3 with reference to statistical analysis against IMR and satellite data will discussed. Here are some examples we plan to add:

*"Experiments were generally comparable when the model results were regionally and monthly averaged. Notable differences were found for the mid-summer nitrate and phosphate concentrations for the Barents, Norwegian and Greenland Seas, as the drawdown of these nutrients was better resolved by EXP1 compared to climatology as EXP1 summer nutrients were lower than in EXP2 and 3. A possible reason why EXP1 has larger drawdown of nutrients during mid-summer is the higher photosynthesis efficiency applied in EXP1 resulting in higher uptake of nutrients and higher zooplankton grazing rate applied to EXP2 and 3 resulting in higher top-down pressure to phytoplankton preventing phytoplankton biomass to uptake more nutrients."*

*"Among the experiments, all perform very similar in terms of nutrient correlations while for nitrate and phosphate, EXP1 performs slightly better in terms of stds, EXP3 performs slightly better for silicate for the Barents Sea and EXP2 for the Norwegian Sea, though the differences among the experiments were almost negligible. Similarly, EXP1 perform better in terms of % bias and RMSE for nitrate and phosphate, and EXP3 perform better for silicate. The slightly better performance of EXP1 for nitrate and phosphate is also evident in summer averages when compared to climatology as mentioned before (Figs. 3a and 3c). The model performance for silicate when using monthly averages shows even less differences among the experiments, however, the EXP3 is slightly closer to WOA timeseries compared to EXP1 (Fig. 3b)."*

*Line 320: Is it not the net primary production your extracting? You don't explicitly model respiration so what you model is gross primary production (photosynthesis) minus respiration i.e. the net?*

We model gross primary production and do not include respiration as a loss term in the equations. Since respiration is not included in the equations, modifying the gross primary production with an arbitrary multiplier to include respiration (e.g. 0.9*GPP) when comparing to observations would add inaccuracy to our results. As the PP observations have a large range, removing 10% PP from the model GPP will still be within the observed PP ranges, thus we settled on providing GPP in the results and noted this in the section with the sentence "Because the model does not have an explicit term for respiration, we can only extract gross primary production from the model, which is then compared to observations. "

*Lines 388-389: What is the reason for the difference in N/P ratios in the Barents sea between WOA and model, I guess mostly as a result of the difference in nitrate in this area?*

The model uses Redfield stoichiometry for production and remineralization in the water column. This strict control on N:P ratios in the organic compartments of the model will restrict dynamical changes in the pelagic N:P ratios. The model's tendency to form a Redfield ratio will be counteracted by the relaxation to open boundary inorganic nutrient conditions of ocean climatology, the river nutrient climatology and the benthic nutrient dynamics.

*Fig. A1: It would be interesting to see a plot comparing ECOSMOII and ECOSMOII(CHL) to observations so as to get a clearer view of the benefits of variable C:Chl ratios.*

We understand that a visual element is necessary to support the statistics table that was included in the manuscript. For this reason, we will add the following figure in the revised manuscript which depicts the data points that were used to calculate the statistics. To distinguish depth layers easily, data points were averaged for each 10 meter depth interval and were depicted with larger markers with color coding for that depth interval.

[Figure]

We will add the following sentences to support this figure: "*While both simulations are statistically similar in general, especially in the deeper euphotic zone (40 – 80 m), ECOSMO II(CHL) statistically performs better at 0 - 20 m and 20-40 m range (Table A1) for almost all statistical quantities. The data that was used to calculate the statistics in Table A1 is visualised in Figure A2 using a scatter plot of the modelled and observed chlorophyll a, which confirms the values in Table A1 showing a slightly better performance of ECOSMO II(CHL) near the surface (0 - 20 m range). 30 m average point is visually slightly better for ECOSMO II, which probably reflects the better % bias performed for 20 – 40 m range (Table A1). While overall the model performance improves, further modifications to either model parameters or formulation should be done for the future iterations of ECOSMO as below 40 meters, the model has not gained a significant improvement suggesting that the chlorophyll a dynamics should be improved for low-light conditions.*"

Other minor comments will be corrected following the referee's suggestions.

Response to Reviewer #3:

*The manuscript describes and evaluates the coupled HYCOM-biogeochemistry ECOSMO II model, configured for the Arctic Ocean. Its evaluation includes some suggestions for fine-tuning of the parameterization according to the geographical characteristics of the expanded domain. The technical description of the ECOSMO II model is well written and constitutes a good baseline for the modeling community. The evaluation of the model results, against observational values, is also of value for a quantitative validation of the model and it is well complemented by the commentary and the figures therein.*

*The paper is very well written, but it would benefit, in my opinion, by a better and more organized description of what observations were used in what experiments.*

Reviewer #1 has also raised this issue and we share the answer here. We will better explain the use of validation data in Section 4.1 opening the section with the following:

'*the model simulations were evaluated using three different datasets, each of which comprises observed inorganic nutrients and chlorophyll a : (1) World Ocean Atlas 2013 (WOA13; Garcia*

*et al., 2013), (2) Institute of Marine Research (2018) data (IMR18), (3) ESA Ocean Colour CCI v5.0 (OC CCI; Ocean Colour Climate Change Initiative; Sathyendranath et al., 2019).'*

This will inform the reader from the start what data is used. In the following paragraphs, we will go through in detail each of them and how and for what purpose they were used will be given. In summary, WOA13 is used for evaluating monthly and regional averages, whereas IMR dataset is used to evaluate the model point-by-point collocating the model to observed data points. The extensive data from IMR18 allowed us to perform the statistical analysis whereas WOA13 was used for a visual comparison. OC CCCI data was used for an additional statistical analysis supporting IMR18 data. To further assist the structure of the observed data, we will add the following figure in the Appendix which includes the profile locations for the IMR data.

[Figure]

*The selection of the areas (and experiments) to show and compare seems ad-hoc. What is the justification for these selections?*

The subregions were defined based on geographical definitions and environmental characteristics such that Barents Sea area defined in this manuscript cover the shelf area south and east of Svalbard, and following the opening to the west (roughly the line we defined, start the deeper ocean that the Norwegian Sea is located. Norwegian Sea is of high interest for our model development and this region in the manuscript extends 20° of latitudes, and such a long latitude coverage can be significant in relation to the timing of the spring bloom, thus in our analyses we divided the region to north and south from halfway. We appreciate you raising concern for a need for a physical setting of the region and added annual temperature climatology to Figure 2 (please see below), and there the reader will see that the Norwegian Sea (N and S) is distinct from the Greenland region where the region borders roughly represent this distinction. Furthermore, the arctic regions (ARC-XXX) were separated from the rest as ARC regions are in general sea-ice covered. You will notice in the figure above that ARC regions are distinctively free of observations, and ARC border separate the observed regions. BERING STR. region is defined to separate it from the rest as it is a

nutrient relaxation to WAO13 region and LAPTEV and KARA are naturally divided with the surrounding islands. SPG region is a subregion within the subpolar gyre area. Necessary text giving this information will be added to the text.

[Figure]

Why not compare all of the areas in all comparisons and statistics? Seems that it wouldn't add much more content and would be of better use in a comprehensive descriptive assessment of the model for this region at large. Perhaps a way to scope the comparative analysis would be to concentrate on one or two of the subregions of the model domain.

For the statistical analysis, the model comparison to IMR data were restricted by data availability, almost exclusive to the NOR.S, NOR.N and BARENTS regions, which are our main area of interest (Please see the figure that was provided with our first comment to Referee #3 for the location of the observations). This figure will be included in the Appendix). For this reason, the manuscript can only focus on these 3 regions (Table 2, Fig.4). For the regional and monthly averaged time-series plots (Fig.3), we were less restricted as we have climatology to compare with. However, this should be done with caution, as in certain regions the climatology is based on very few observations. To show this, we will add a figure (Fig. A2) that depicts the number of observed points for each region. After our inspection on the number of samples, we decided to include the regions of meaningful coverage in Figure 3, NOR.N, NOR.S, BARENTS, GREENLAND, SPG and KARA. We note that KARA is also limited in data outside summer period. BERING STR is not included in the figure as it is too close to the relaxation zone of the model. The text in the manuscript is extensively modified to account for these changes.

[Figure]

The performance comparison for the model would be well prescribed in a table of EXPT vs. AREA with some objectively-derived performance metric value (i.e. rmse/skill score).

As pointed out above, we can only do the statistical analysis for the regions where we have data and these regions were already covered in the text. Statistics (rmse, bias, correlation and standard deviations for nitrate, silicate, phosphate and chlorophyll a) for EXPT and AREA are given in Table 2. We tried to further extend the chlorophyll error statistics with the satellite data but for this region, satellite chlorophyll is hindered by clouds, ice cover and dark seasons. Therefore, we kept the statistical analysis in Table 4 and 5, Figure 9 consistent with the regions in Table 2 and Figure 3. But following the comments from the other referees, the text will be extensively modified to include results for EXP1, EXP2 and EXP3 and comment on each of their strengths. The modification to Figure 3 now depicts both Norwegian regions, Greenland and Kara Sea which would assist further towards your concerns.

This would allow for a good summary of the many combinations of experiments and areas covered. The many regions, very comprehensive, somewhat dilutes the detailed validation of the model itself, only providing a general descriptive validation with some statistical quantification for some of the areas.

We value the necessity of domain wide analysis as you also suggested here and above, and for this reason we included domain wide model output (Figs. 5 and 8) with region specific model variable quantities for the readers interest and reference for future studies.

While technically, this is a paper of excellent value, scientifically, it could have been further elaborated to elucidate what the model brings in terms of better understanding of the hydrodynamics and biogeochemistry of the Arctic region. It would also benefit from a more detailed description, via equations, diagrams, and text, of the plankton group dynamics and again, the scientific value of those results.

*In summary, the paper is very well written and the technical topic is well covered, but the scientific value of the paper could be improved with more details on the model parameterization, inherent errors therein, and the "take home" message of what is learned scientifically.*

As a model description paper, the main focus of this study is to provide the technical changes to ECOSMO, present the final state of the model, and evaluate the model performance using different parameterizations. While doing so, we have long-term model time-series spanning for 2 decades for the Nordic Seas and the Arctic. We included the general biogeochemistry of this region within a technical paper context. We plan for further scientific investigations of this data in future studies. However, considering the referee's comments, we will revise the manuscript towards strengthening its existing scientific messages in the abstract, conclusion and Section 5 will have improved scientific emphasis. Specifically:

- Comment on the value added to the model by the including explicit chlorophyll a
- What do different parameter values mean in terms of model performance and internal dynamics.
- Lessons learned with respect to future evolutions of the model

Reviewer #1 also commented on expanding the equations in the manuscript and we will follow both the suggestions of Reviewer #1 and #3 and detailed descriptions of phytoplankton growth dynamics will be included. Please refer to our answer to Reviewer #1 on the subject.

Other minor comments will be corrected following the referee's suggestions.

---

## Author Response (AR1)

Dear Editor,

Please find our revised manuscript based on the referees' comments provided during the discussion phase of the submission process. You will find our point-by-point responses to every comment made by the referees. The revised manuscript, including the 'track changes', is included following our responses to the referees. We wish to thank the referees for their insightful comments that have contributed to improving the manuscript.

Our changes to the manuscript include:

- Revising manuscript to objectively analyse each of the experiment, and document the model performance for the users of each of these experiments (i.e. parameterization sets) as these experiments represent active use cases of ECOSMO. We have improved the clarity of the technical description and scientific messages of the manuscript.
- The text regarding sources and use of data for model evaluation has been revised to make it clear for the reader where the data comes from and for what purpose it was used. As can be seen in the revised manuscript, each dataset (climatology, in situ and satellite) has its own value especially with regard to spatial and temporal coverage and they complement each other well for model evaluation.
- The text has been improved in its scientific content with the following key points:

    o Comment on the added value of using explicit state variables for chlorophyll a in the model formulation
    o What do different parameter values mean in terms of model performance and internal dynamics.
    o Lessons learned with respect to future evolutions of the model

We have carefully considered all the technical points raised by the reviewers and made the necessary changes to the text. To improve the clarity of the manuscript and enhance its technical and scientific points, minor changes to the text were made. Any changes to the manuscript can be seen in detail with the 'track changes' enabled below following our responses to the referees.

**Reply to comments from the reviewers**

**Reviewer #1**
Referee's comments are provided in 'black italic', and our responses in 'blue'. Our proposed changes to the text appear in 'green'. We reference the manuscript provided below with its line numbers (e.g. L10-L19).

*The paper presents and evaluates an updated version of the marine biogeochemical model ECOSMO II, that now also includes a parameterisation for a variable C:Chl ratio of phytoplankton. The biogeochemical is coupled to the HYCOM ocean model, configured for the Arctic Ocean. Model evaluation of a default setup is carried out against observed nutrients and Chl a from in situ measurements and remote sensing, and complemented by comparison to literature values of primary production. Model sensitivity to biogeochemical parameters is*

*evaluated in two further experiments. In general, all model configurations perform quite well with regard to the observations, and improvement in surface chlorophyll when parameters are adapted to the hydrodynamics of the Arctic.*

*In general, the paper is well written and I very much appreciate the thorough comparison to observed tracers. One drawback is the - to my eyes - somewhat incomplete description of phytoplankton (see (1) below), and the slightly confusing description of how observations were used for comparison (2). I further encourage the authors to improve on the structure of the paper (3) and to extend a bit on the comparison of the three different experiments (see my comment no 4 below).*

*(1) For better understanding of how phytoplankton responds to nutrient and light limitation I suggest to present the equation(s) for phi (in Eqns 1 and 2). Without knowledge about this functional form it is difficult for the reader to understand its dynamics, in particular as DS2013 might not be accessible to everyone. Further, does the C:Chl ratio (or the amount of chlorophyll in phytoplankton) affect the light attenuation and P-I curve, i.e. the of phytoplankton growth rate?*

We agree that the term 'phi' (growth limitation) is necessary to be present in a model description paper and understand the referee's concern that the DS2013 paper may not be accessible for everyone. For this reason, we include $\phi_{P_j} = \min\left(\alpha(I), \beta_N, \beta_P, \beta_{Si}\right)$ as the general equation for phi and expand this equation to include all of the necessary formulations for each individual term in the equation totalling to additional 6 equations. We believe these additions will address the request of the referee. The changes are as follows and each equation define a limitation respective to light or nutrient:

**(L131-139)**

$$\phi_{P_j} = \min\left(\alpha(I), \beta_N, \beta_P, \beta_{Si}\right) \tag{3}$$

$$\alpha(I) = \tanh\left(\varphi I(x, y, z, t)\right) \tag{4}$$

$$\beta_N = \beta_{NH_4} + \beta_{NO_3} \tag{5}$$

$$\beta_{NH_4} = NH_4/(NH_4 + r_{NH_4}) \tag{6}$$

$$\beta_{NO_3} = (NO_3/(NO_3 + r_{NO_3}))\exp\left(-\gamma NH_4\right) \tag{7}$$

$$\beta_{PO_4} = PO_4/(PO_4 + r_{PO_4}) \tag{8}$$

$$\beta_{Si} = Si/(Si + r_{Si}) \tag{9}$$

And the definition of these terms in the text:

**(L146-148)** DS2013 give $\sigma_j$, $\phi_{P_j}$, $\varphi$, $r_{NH_4,NO_3,PO_4,Si}$, $\gamma$, $G_i$ and $m_{P_j}$ as the phytoplankton maximum growth rate, growth limitation, photosynthesis efficiency parameter, nutrient-specific half saturation constant, $NH_4$ inhibition parameter, zooplankton grazing rates and mortality rates respectively.

For the reader to follow better, PAR was removed from the equations and was given as I(x, y, z, t) suggesting a point in 3D and time. PAR is dependent on surface radiation, $I_S$, water and chlorophyll specific attenuation constants and CHL concentration. C:Chl ratio indirectly affects the light attenuation as Chl is included there. We believe the equation in this form better relates the reader with CHL vs light attenuation. The changes to the text are as follows:

**(L130)** $\rho_{chl_j} = \dfrac{\theta_P^{max}\phi_{P_j}C_{P_j}}{\alpha_{P_j}I(x,y,z,t)\,Chl}$ (1)

**(L146)** Photosynthetically active radiation (PAR) is given as I(x,y,z,t).

**(L154-155)** In relation to the addition of a prognostic chlorophyll *a* state variable, photosynthetically active radiation I(x,y,z,t) at depth undergoing attenuation was modified to have chlorophyll *a* in the exponential term:

**(L157)** $I(x,y,z,t) = \dfrac{I_s(x,y)}{2}exp\left(-k_w z - k_{Chl}\int_z^0 \sum_{j=1}^2 Chl_{P_j}\partial z\right)$ (12)

The model does not relate CHL to P-I curve at the moment in the context of growth, but it is a very valid point and will be including such relations in future iterations of ECOSMO. We have noted this in the discussion sections:

**(L594-599)** An important addition to explicit chlorophyll *a* variable is the inclusion of the initial slope of P-I curves to light-limitation on growth. In this study, light-limitation on growth was approached in a PFT and chlorophyll *a* independent fashion (Eq. 4). Future versions of ECOSMO should adopt ways (e.g. Evans and Parslow, 1985) to include the PFT specific P-I curve slopes to take full advantage of the explicit chlorophyll a variable. This would allow PFTs to differentiate their niche light conditions for production, and further allow better integration with the bio-optical modelling of the marine environment.

*(2) The description of type of observations, and how these were used for model evaluation (section 4.1) is somewhat confusing (see also below, specific comments).*

Also

*Line 220-221: "Nitrate, silicate, phosphate and chlorophyll a in situ data from Institute of Marine Research (2018) were used for the statistical evaluation of the model results." Where were these data used and what is the difference to the comparison against WOA2013 data?*

Reviewer-3 has also raised this issue. We explain the use of validation data better in Section 4.1 opening the section with the following:

**(L257-259)** The model simulations were evaluated using three different datasets as follows: (1) World Ocean Atlas 2013 (WOA13; Garcia et al., 2013), (2) Institute of Marine Research (2018) data (IMR18), (3) ESA Ocean Colour CCI v5.0 (OC CCI; Ocean Colour Climate Change Initiative; Sathyendranath et al., 2019).

That will inform the reader from the start what data is used. In the following paragraphs, we go through each of them in detail and explain how and for what purpose they were used. In summary, WOA13 is used for evaluating monthly and regional averages, whereas the IMR dataset is used to evaluate the model point-by-point collocating model against in situ data points. The extensive data from IMR18 allowed us to perform the statistical analysis whereas WOA13 was used for a visual comparison. OC CCCI data was used for an additional statistical analysis supporting IMR18 data.

The following sentences from the text should give the reader enough information on the type of data, and what it was used for:

(L261-262; WOA data) The model's consistency with the large-scale climatological inorganic nutrient distributions was quantified by comparing the regionally averaged monthly inorganic nutrient model data (nitrate, silicate and phosphate) to WOA13 data.

(L267-269; WOA data) These monthly time-series allowed a model evaluation for the regions  , in which the in situ data was not optimal for the statistical analysis. Regional climatology data should be used with caution because WOA13 data are in some places based on very few observations and that may mislead the evaluation process.

(L274-278; IMR data) A separate evaluation for the model inorganic nutrients (nitrate, silicate and phosphate) and chlorophyll *a* was conducted using the IMR18 in situ data by performing a point-by-point (location and depth) co-location for the statistical analysis.

(L285-288; OC CCI data) A final model chlorophyll evaluation was conducted using OC CCI daily surface chlorophyll *a* and downwelling attenuation coefficient at 490 nm (kd490) at 4km x 4km spatial resolution.

(L294-296; OC CCI data) This separate analysis allows us to include chlorophyll *a* data for model evaluation in addition to IMR18 data.

To further assist the structure of the observed data, we added Figure A5 in the Appendix which includes the profile locations for the IMR data. (L698)

We would like to answer item 3, 4 and some of the minor comments in the following together.

*(3) Some sentences of the model setup (section 3) already anticipate the results (e.g., lines 183-186: "Using lower ... column stabilizes." and 189-191: During the continuous ... will replace the parameterization set in EXP2."). I would suggest to reorganise the paper a bit (see also comment (4)), and draw conclusions only from the model evaluation.*

Sentences anticipating the results were removed. Please see the crossed out sentences in L212-225. Please also refer to our extensive comments on the subject below.

*(4) I enjoyed reading the analysis in section 4.3. I think this analysis could be complemented by constrasting it with the results obtained with EXP2 and EXP3. For example, Table 2 shows that EXP1 always performs best with regard to the correlation coefficient, in 7 (8) cases out of 12 with regard to the normalised StdDev (Bias%), and in half of the cases with regard to RMSE. Likewise, it performs always best (regardless of metric) for phosphate and nitrate, and in half of the cases with regard to chlorophyll. Even if some differences to EXP2 and EXP3 are only small, I think this could be discussed a bit more (so far, there seems only be a sentence in line 451), and contrasted with the outcome presented in Tables 4 and 5, which indicate that EXP3 performs best with*

*regard to surface Chl a. What is the reason for these differences? Is it because Table 4 and 5 only refer to surface values?*

*Line 278: "0.6-0.72" - Table 2 gives 0.79-0.83 for EXP1: confusion with EXP3?*

*Line 279ff: For the biases Table 2 gives the relative bias, but in the text you refer to absolute biases, which is somewhat confusing.*

*Line 280: "2.47-3.34" - this seems to be the bias of EXP3. Line 290: "0.74-0.78" - again, for EXP3?*

*Line 300: "0.81-0.89" - again EXP3?*

*Line 311: "0.97-1.2" - EXP3?*

*Line 451: "EXPeriment statistics for inorganic nutrients are very similar in all experiments (Table 2)." - Overall, to me it seems as if EXP1 performs better than EXP2 and EXP3 (see above my comment no. (4))*

Referee #2 has also raised similar concerns. We agree that our review of the model results and discussions on the statistics of different experiments were misleading. Therefore, the text on Section 4.3 was extensively reviewed. Going through the comments and the manuscript, we understand how the text reads to be in favor of EXP3 which was not our intent. We value all the experiments as the parameter sets in these experiments are being actively used, and our intention in this manuscript was to document the model performance for each experiment. For this reason, along with many other revisions in the manuscript, we clearly state the use cases for each experiment in the introduction and the model setup, we reference what each experiment represents (i.e. EXP1 for the original ECOSMO, EXP2 for the operational model prior to July 2021 and EXP3 for the current operational model after July 2021).

To achieve an objective comparison among the experiments, the text regarding the statistics to which the referee refers now include results from all the experiments, not only EXP1, which also corrects the confusion stemming from EXP1 vs EXP3 as stated by the referee, as such each statistical range given in the text covers the min/max of all experiments (EXP1, 2, and 3). We have also noted that EXP1 better performs well in many aspects and each of these are documented in the text for both the statistical analysis and the monthly average comparison to WOA (as we also note in the text that EXP1 averages perform better for nitrate and phosphate) and discuss possible reasons. EXP2 is also included in these new additions.

The contrast between performances among EXP1 and EXP3 with reference to statistical analysis against IMR and satellite data are discussed better. Here are the changes, with their respective sections indicated:

**Abstract:**

**(L16-17)** We document the performance of each parameter set objectively analysing the experiments against in situ, satellite and climatology data.

The sentences favoring EXP3, which leads to confusion were removed **(L24-28)**.

**Introduction:**

**(L78-81)** We present the results from three experiments using ECOSMO II(CHL) adopting different parameter sets from DS2013 (the original parameter set tuned for the North and Baltic Seas), CMEMS Artic operational model prior to June 2021 and the current Arctic operational model parameterization.

The sentences which reads as the manuscript focuses only on EXP3 were removed **(L77-78 and L81-84)**.

[revised manuscript text omitted]

*Line 36: Here I would prefer a more specific link to a model application at copernicus (I had to search around a bit).*

Added doi fort he ARC MFC model. **(L-39)**

*Line 89: 'use the time stepping' – What time step lenghts were applied?*

20 minutes. Added to the text. **(L97)**

*Line 172: Evaluation after just two years of spinup seems to be quite soon - do you have any indication if the model drift (with regard to the BGC components) has decreased to some specific value?*

Below we provide model average (spatial and vertical) evolution of nutrients between 1991 and end of 2010 during which the statistical analyses were performed. Inspecting the first years, the model does not have any abnormal drifts that could be interpreted as a the spinup prior to the analyses were sufficient. We note that, unless the referee or the editor have objections, we will not include the following figures in the manuscript.

[Figure]

[Figure]

*Line 201-202: "The dynamics shown in Appendix A1 is expected to be valid for each model point." - How is this expectation justified?*

The sentence will be changed with the following:

**(L233-234)** The dynamics shown in Appendix A1 is representative for regions with similar plankton dynamics (e.g. Norwegian Sea, Barents Sea), thus can be used as a showcase for the new chlorophyll *a* specific addition.

*Line 229 "Processed model chlorophyll" - What does "processed" mean in this context?*

We were aiming for the meaning 'post-processed', but the sentence is modified to:

**(L289-291)** We used this dataset for the years 1998-2010. Chlorophyll *a* and kd490 were remapped to the model grid and the model chlorophyll *a* was averaged within 1/kd490 (m) depth.

*Figure 3 caption and elsewhere: "WOA18" - above you refer to WOA2013*

The text is made consistent throughout and states WOA13 as the model evaluation is performed using the 2013 version. The one exception is the 'temperature' depicted in Figure 2 **(L175)** is WOA 2018. This was a recent addition based on Referee #3's comments.

*Line 383-383: "silicate concentrations ... in the climatological data." - are discharge rates into the Arctic very different for different types of nutrients? (I.e., outside the assumed stoichometry)? Perhaps a note on this would be interesting ...*

We added:

**(L515-519)** As the model is relaxed towards the climatology at the Bering Strait through a sponge layer in the model domain, the overall high nutrient concentrations near the Bering Strait and especially the high silicate concentrations at the Siberian coast due to higher Si/N ratio of Pacific origin water masses compared to the Atlantic water masses, and the addition of high Si/N ratio river discharge is reflected in the modelled annual averages.

*Lines 443-446: It is not really clear to me what you want to say with this sentence - could it be rephrased?*

The original text was:

EXP1 perform a very fast phytoplankton response to light availability during the spring bloom period where light availability is increased resulting in a steep curve where chlorophyll a concentrations are notably higher compared to the observations (results not shown) evident in the high %biases, whereas EXP2 and EXP3 have closer concentrations during spring bloom.

It was rephrased to:

**(L572-575)** The consistent higher bias of EXP1 compared EXP2 and EXP3 can be explained by its higher photosynthesis efficiency (Table 1). EXP1 show a very fast primary production response to light availability during the spring bloom period with notably higher chlorophyll *a* concentrations compared to the observations (results not

shown) evident in the high %biases, whereas chlorophyll *a* concentrations in EXP2 and EXP3 are closer to observed values during spring bloom.

*Lines 478-479: "using realistic ... spin-up period" - what are the realistic constant values? and: constant with respect to what - over time? over depth?*

We agree that the sentence is misleading. The original text was:

We performed a 27-year run starting in 1990 using realistic constant values for the biogeochemical variables and considered the first 5 years as the spin-up period.

We corrected it with:

**(L660-661)** We performed a 27-year run starting in 1990 using WOA2013 profiles from January climatology for the biogeochemical variables and considered the first 5 years as the spin-up period.

Response to Reviewer #2:

Referee's comments are provided in 'black italic', and our responses in 'blue'. Our proposed changes to the text appear in 'green'. We reference the manuscript provided below with its line numbers (e.g. L10-L19).

*The paper presents a description and evaluation of a new configuration of the biogeochemical model ECOSMO II (ECOSMO II(CHL)) in which they have included chl in the three functional phytoplankton types as prognostic variables. Furthermore, the paper presents a comparison of model experiments using three different parameter sets. In the appendix, the authors have also included a comparison between ECOSMO II and ECOSMO II(CHL) in a configuration where the models are coupled to the 1d physical model GOTM.*

*The paper is well written and the methodology ambitious. However, I would like to have seen a larger focus on the comparison between ECOSMO II and ECOSMO II(CHL). Furthermore, apart from two sentences (451-453) I find no discussion on why EXP1 seems to perform better*

*for nitrate and phosphate (Fig 3 and Table 2) than EXP 2 and 3. I find that the paper could be published in GMD after some minor revisions.*

*Lines 139-144. What limited the concentrations from becoming too small before you added this? Also, you state that: "The minimum concentration at which the loss terms are calculated are...". Should it say: " The minimum concentration at which the loss terms are switched off"?*

There was no limit before the addition of the on/off switch, which quickly became a problem for a well-mixed open ocean. Hence we added this modification. It was a necessity for preventing a very late spring bloom, as the biomasses reached almost 0 after every winter. We agree with your suggestion and rephrased the sentence accordingly:

**(L166-168)** The minimum concentration at which the loss terms are switched off are 0.1, 0.005 and 0.01 mgC m$^{-3}$ for phytoplankton, chlorophyll *a* and zooplankton respectively.

*Fig.3: It is not obvious to me that EXP2 and 3 perform better than EXP1. It looks the opposite from this figure.*

Referee #1 has also raised similar concerns. We would like to share the same comment here. For exact changes in the text, please refer to the comment we made to Referee #1's comments (3) and (4).

We agree that our review of the model results and discussions on the statistics of different experiments were misleading. Therefore, the text on Section 4.3 was extensively reviewed. Going through the comments and the manuscript, we understand how the text reads to be in favor of EXP3 which was not our intent. We value all the experiments as the parameter sets in these experiments are being actively used, and our intention in this manuscript was to document the model performance for each experiment. For this reason, along with many other revisions in the manuscript, we clearly state the use cases for each experiment in the introduction and the model setup, we reference what each experiment represents (i.e. EXP1 for the original ECOSMO, EXP2 for the operational model prior to July 2021 and EXP3 for the current operational model after July 2021).

To achieve an objective comparison among the experiments, the text regarding the statistics to which the referee refers now include results from all the experiments, not only EXP1, which also corrects the confusion stemming from EXP1 vs EXP3 as stated by the referee, as such each statistical range given in the text covers the min/max of all experiments (EXP1, 2, and 3). We have also noted the EXP1 better performs well in many aspects and each of these are documented in the text for both the statistical analysis and the monthly average comparison to WOA (as we also note in the text that EXP1 averages perform better for nitrate and phosphate) and discuss possible reasons. EXP2 is also included in these new additions.

The contrast between performances among EXP1 and EXP3 with reference to statistical analysis against IMR and satellite data are discussed better.

*Line 320: Is it not the net primary production your extracting? You don't explicitly model respiration so what you model is gross primary production (photosynthesis) minus respiration i.e. the net?*

We model gross primary production and do not include respiration as a loss term in the equations. Since respiration is not included in the equations, modifying the gross primary production with an arbitrary multiplier to include respiration (e.g. 0.9*GPP) when comparing to observations would add inaccuracy to our results. As the PP observations have a large range, removing 10% PP from the model GPP will still be within the observed PP ranges, thus we settled on providing GPP in the results and noted this in the section with the sentence **(L436-437)** "Because the model does not have an explicit term for respiration, we can only extract gross primary production from the model, which is then compared to observations. "

*Lines 388-389: What is the reason for the difference in N/P ratios in the Barents sea between WOA and model, I guess mostly as a result of the difference in nitrate in this area?*

The model uses Redfield stoichiometry for production and remineralization in the water column. This strict control on N:P ratios in the organic compartments of the model will restrict dynamical changes in the pelagic N:P ratios. The model's tendency to form a Redfield ratio will be counteracted by the relaxation to open boundary inorganic nutrient conditions of ocean climatology, the river nutrient climatology and the benthic nutrient dynamics.

*Fig. A1: It would be interesting to see a plot comparing ECOSMOII and ECOSMOII(CHL) to observations so as to get a clearer view of the benefits of variable C:Chl ratios.*

We understand that a visual element is necessary to support the statistics table that was included in the manuscript. For this reason, we added a figure (L698) in the revised manuscript which depicts the data points that were used to calculate the statistics. To distinguish depth layers easily, data points were averaged for each 10 meter depth interval and were depicted with larger markers with color coding for that depth interval.

We added the following text to support the figure:

**(L677-685)** While both simulations are statistically similar in general, especially in the deeper euphotic zone (40 – 80 m), ECOSMO II(CHL) statistically performs better at 0 - 20 m and 20-40 m range (Table A1) for almost all statistical quantities. The data that was used to calculate the statistics in Table A1 is visualised in Figure A2 using a scatter plot of the modelled and observed chlorophyll *a*, which confirms the values in Table A1 showing a slightly better performance of ECOSMO II(CHL) near the surface (0 - 20 m range). 30 m average point is visually slightly better for ECOSMO II, which probably reflects the better % bias performed for 20 – 40 m range (Table A1). While overall the model performance improves, further modifications to either model parameters or formulation should be done for the future iterations of ECOSMO as below 40 meters, the model has not gained a significant improvement suggesting that the chlorophyll *a* dynamics should be improved for low-light conditions.

Response to Reviewer #3:

Referee's comments are provided in 'black italic', and our responses in 'blue'. Our proposed changes to the text appear in 'green'. We reference the manuscript provided below with its line numbers (e.g. L10-L19).

*The manuscript describes and evaluates the coupled HYCOM-biogeochemistry ECOSMO II model, configured for the Arctic Ocean. Its evaluation includes some suggestions for fine- tuning of the parameterization according to the geographical characteristics of the expanded domain. The technical description of the ECOSMO II model is well written and constitutes a good baseline for the modeling community. The evaluation of the model results, against observational values, is also of value for a quantitative validation of the model and it is well complemented by the commentary and the figures therein.*

*The paper is very well written, but it would benefit, in my opinion, by a better and more organized description of what observations were used in what experiments.*

Reviewer #1 has also raised this issue. We would like to share the same comment here. For exact changes in the text, please refer to the comment we made to Referee #1's comments (2).

In summary, WOA13 is used for evaluating monthly and regional averages, whereas the IMR dataset is used to evaluate the model point-by-point collocating model against in situ data points. The extensive data from IMR18 allowed us to perform the statistical analysis whereas WOA13 was used for a visual comparison. OC CCCI data was used for an additional statistical analysis supporting IMR18 data.

*The selection of the areas (and experiments) to show and compare seems ad-hoc. What is the justification for these selections?*

The subregions were defined based on geographical definitions and environmental characteristics such that Barents Sea area defined in this manuscript cover the shelf area south and east of Svalbard, and following the opening to the west (roughly the line we defined, start the deeper ocean that the Norwegian Sea is located. Norwegian Sea is of high interest for our model development and this region in the manuscript extends $20^o$ of latitudes, and such a long latitude coverage can be significant in relation to the timing of the spring bloom, thus in our analyses we divided the region to north and south from halfway. We appreciate you raising concern for a need for a physical setting of the region and added annual temperature climatology to Figure 2 **(L175)**, and there the reader will see that the Norwegian Sea (N and S) is distinct from the Greenland region where the region borders roughly represent this distinction. Furthermore, the arctic regions (ARC-XXX) were separated from the rest as ARC regions are in general sea-ice covered. You will notice in the figures **(L708 and L715)** that ARC regions are distinctively free of observations, and ARC border separate the observed regions. BERING STR. region is defined to separate it from the rest as it is a nutrient relaxation to WAO13 region and LAPTEV and KARA are naturally divided with the surrounding islands. SPG region is a subregion within the subpolar gyre area. Necessary text for this information is added to the text.

**(L248-255)** The extent of the subdomains depicted in Figure 2 were defined by the geographical definitions of the regions and their environmental setting as such the BARENTS region covers the shelf area south and east of Svalbard and border NOR. N at the opening to the Norwegian Sea where it is deeper and is highly influenced by the Atlantic inflow. The Norwegian Sea is divided into north and south to take into account for the differences in daylength across the wide latitude range ($20^o$). The border between GREENLAND and NOR. N and S. roughly locates the temperature changes of the different water masses in the region (Fig. 2). ARC regions were set to cover sea-ice covered regions most of the year. BERING STR. Region was set to separate the boundary conditions from the rest of the domain. KARA and LAPTEV regions have naturally defined borders with the islands around them. SPG region is defined to represent the subpolar gyre region.

*Why not compare all of the areas in all comparisons and statistics? Seems that it wouldn't add much more content and would be of better use in a comprehensive descriptive assessment of the model for this region at large. Perhaps a way to scope the comparative analysis would be to concentrate on one or two of the subregions of the model domain.*

For the statistical analysis, the model comparison to IMR data were restricted by data availability, almost exclusive to the NOR.S, NOR.N and BARENTS regions, which are our main area of interest (Please see the figure that was added after the referees' comments for the location of the observations (Fig. A5; L715)). For this reason, the manuscript regarding statistics against in situ data can only focus on these 3 regions (Table 2, Fig.4).

For the regional and monthly averaged time-series plots (Fig.3; L359), we were less restricted as we have climatology to compare with. However, this should be done with caution, as in certain regions the climatology is

based on very few observations. To show this, we added a figure (Fig A3; L708) that depicts the number of observed points for each region. After our inspection on the number of samples, we decided to include the regions of meaningful coverage in Figure 3, NOR.N, NOR.S, BARENTS, GREENLAND, SPG and KARA. We note that KARA is also limited in data outside summer period. BERING STR is not included in the figure as it is too close to the relaxation zone of the model. The text in the manuscript is extensively modified to account for these changes. Note that Figure 3 (L359) was updated after Referee #3's comments to extend the analysis coverage.

*The performance comparison for the model would be well prescribed in a table of EXPT vs. AREA with some objectively-derived performance metric value (i.e. rmse/skill score).*

As pointed out above, we can only do the statistical analysis for the regions where we have data and these regions were already covered in the text. Statistics (rmse, bias, correlation and standard deviations for nitrate, silicate, phosphate and chlorophyll a) for EXPT and AREA are given in Table 2. We tried to further extend the chlorophyll error statistics with the satellite data but for this region, satellite chlorophyll is hindered by clouds, ice cover and dark seasons. Therefore, we kept the statistical analysis in Table 4 and 5, Figure 9 consistent with the regions in Table 2 and Figure 3. But following the comments from the other referees, the text was extensively modified to include results for EXP1, EXP2 and EXP3 and comment on each of their strengths. Please refer to our comments to Referee #1 items (3) and (4).

In this regard, here are additions to the text:

**(L268-272)** Regional climatology data should be used with caution because WOA13 data are in some places based on very few observations and that may mislead the evaluation process. To detect the regions with low number of observations, WOA13 data points were extracted for each region and were summed up as monthly time-series (Fig. A3). As an example, the number of data points for the regions defined as ARC (Fig. 2) were almost negligible compared to the Norwegian Sea or the Barents Sea. Further discussion on this is given in Section 4.3.

**(L325-330)** We note that the number of samples for the monthly climatology vary between months and regions (Fig. A3). Especially for the cases of polar regions and eastern coastal Arctic, the number of data points that were used to construct the monthly climatology were negligible compared to the remaining southern regions (Fig. 2). We have also included KARA region in the discussion here as there are significant number of data points, though limited to only late-summer (months 7 – 11). Even in the case of the Norwegian and the Barents Seas, the number of samples for winter months are significantly lower than the rest of the year.

**(L346-358)** The Kara Sea is highly influenced by the coastal nutrient discharges as can be seen from the high standard deviations, especially for silicate including the late-summer where we have sufficient data for this analysis. Apart from surface silicate, the model performs generally well for the Kara Sea from month 7 and onwards. In addition to our comments about silicate above, the coastal discharge of nutrients should be improved in future studies, as in this study we used annual climatology for river nutrient discharge.

Experiments were generally comparable when the model results were regionally and monthly averaged (Fig. 3). Notable differences were found for the mid-summer nitrate and phosphate concentrations for the Barents, Norwegian and Greenland Seas, as the drawdown of these nutrients was better resolved by EXP1 compared to climatology as EXP1 summertime nutrients were lower than in EXP2 and 3. A possible reason why EXP1 has larger drawdown of nutrients during mid-summer is the higher photosynthesis efficiency applied in EXP1 resulting

in higher uptake of nutrients and higher zooplankton grazing rate applied to EXP2 and 3 resulting in higher top-down pressure to phytoplankton preventing phytoplankton from consuming more nutrients.

*This would allow for a good summary of the many combinations of experiments and areas covered. The many regions, very comprehensive, somewhat dilutes the detailed validation of the model itself, only providing a general descriptive validation with some statistical quantification for some of the areas.*

We value the necessity of domain wide analysis as you also suggested here and above, and for this reason we originally included domain wide model output (Figs. 5 and 8) with region specific model variable quantities for the readers interest and reference for future studies.

*While technically, this is a paper of excellent value, scientifically, it could have been further elaborated to elucidate what the model brings in terms of better understanding of the hydrodynamics and biogeochemistry of the Arctic region. It would also benefit from a more detailed description, via equations, diagrams, and text, of the plankton group dynamics and again, the scientific value of those results.*

*In summary, the paper is very well written and the technical topic is well covered, but the scientific value of the paper could be improved with more details on the model parameterization, inherent errors therein, and the "take home" message of what is learned scientifically.*

As a model description paper, the main focus of this study is to provide the technical changes to ECOSMO, present the final state of the model, and evaluate the model performance using different parameterizations. While doing so, we have long-term model time-series spanning for 2 decades for the Nordic Seas and the Arctic. We included the general biogeochemistry of this region within a technical paper context. We plan for further scientific investigations of this data in future studies. However, considering the referee's comments, we will revised the manuscript towards strengthening its existing scientific messages in the abstract, conclusion and Section 6 has improved scientific emphasis. Specifically:

- Comment on the value added to the model by the including explicit chlorophyll a
- What do different parameter values mean in terms of model performance and internal dynamics.
- Lessons learned with respect to future evolutions of the model

Reviewer #1 also commented on expanding the equations in the manuscript and we followed both the suggestions of Reviewer #1 and #3 and detailed descriptions of phytoplankton growth dynamics were included. Please refer to our answer to Reviewer #1 on the subject.

Specific addditions to the text are:

[revised manuscript text omitted]
 While WOA18 data was used to evaluate model monthly averages, model Nnitrate, silicate, phosphate and chlorophyll a was compared by performing a point-by-point (location and depth) to in situ data from Institute of Marine Research (2018)co-locationomparison were used for the statistical evaluation of the model resultsanalysis (cf. Section 4.3; Table 2). For each in situ data point, the closest model grid was selected and the vertical profile was interpolated to the observed depth. Data with only 'good' flags were used totalling to more than 120000 data points for each nutrient and chlorophyll *a*.

280  While the size of the observed dataset is unique, the regional coverage is limited to mainly the Norwegian Sea and the Barents Sea (Fig. A5). For this reason, the analysis using WOA13 and IMR18 complement each other well with one covering wider regions and the other providing a large dataset respectively.

Very few direct observations of primary production are available in our focus region. We have therefore used reported values from the literature for evaluating the estimated magnitude of primary production (cf. Sect. 5 for the references). A final model chlorophyll evaluation was conducted using OC CCI ESA Ocean Colour CCI v5.0 (Ocean Colour Climate Change Initiative; Sathyendranath et al., 2019) daily surface chlorophyll *a* and downwelling attenuation coefficient at 490 nm (kd490) at 4km x 4km spatial resolution were used for evaluation of simulated chlorophyll a. This dataset is derived from multiple sensors: SeaWIFS, MODIS Aqua, MERIS, SeaWIFS LAC and VIIR. We used this dataset for the years 1998-2010. Chlorophyll *a* and kd490 were remapped to the model grid and the model chlorophyll *a* was processed by averaginged within thetowithin 1/kd490 (m) depth. In the cases that kd490 data were missing, 1/kd490 valuethe model chlorophyll *a* was set toaveraged within 10 meters. Processed mModel chlorophyll *a* was then statistically analyzedanalysed 
[revised manuscript text omitted]

---

## Author Response (AR2)

Dear Dr. Yool,

Thank you very much for your positive response to the revised manuscript and your further constructive comments. Below you will find our replies to your comments, along with their exact location in the tracked document that is uploaded together with the revised manuscript.

I have examined the new manuscript version and am generally satisfied that it addresses the issues raised by your referees. I have listed a few minor points below that would benefit from some additional clarification.

• Please include DOIs or weblinks (with access dates) to the observational datasets used in your analysis
We have included a new section in the Appendix that lists the weblinks of the observational datasets that were used to evaluate the model. We used your approach in the link you provided as a guide to structure the section. (L652-664)

• It would be useful if something of the response you gave around spin-up somehow appeared in the manuscript; for instance, quantifying the linear rate of change in annual mean nutrients, either as absolute or relative (%) numbers
We understand the necessity and as you suggest we have added absolute rates of changes to the manuscript. The following paragraph is added to the beginning of the model evaluation section (L287-291):

*Prior to presenting detailed model results, we note that the model is at a steady-state after 2-years of simulation (1989 – 1990). For the years 1991 – 2010, which we have performed our analyses, the model nitrate rate of change is 0.002, 0.0031 and -0.0007 mmolN $m^{-3}$ $y^{-1}$ for average nitrate within 0 – 100 m, 0 – 500 m, and 0 m – bottom respectively. For the same averaging depths, the model silicate and phosphate rates of change are 0.004, 0.0055, 0.0057 mmolSi $m^{-3}$ $y^{-1}$ and 0.00004, 0.00016, -0.0026 mmolP $m^{-3}$ $y^{-1}$ respectively.*

• In your original manuscript there was some confusion around WOA2013 and WOA2018; the revised manuscript clarifies this, but I note a mention of WOA2018 on line 629.
We have corrected the WOA version to 2013. We apologize for this as it should have been corrected in the revised manuscript.

Further, I did identify some issues when revisiting your manuscript in relation to the "code and data availability" section. This has changed since the previous manuscript revision, and no longer makes clear the DOI link to Zenodo, something which is not clear from the tracked changes version of the revised manuscript. Please revert this to restore direct mention of the archive.
We reverted back to an earlier version of this section. You have approved that version before. We believe this clarifies the reference to Zenodo (L668-680).

Additionally, while the description of the archive's contents is extensive in this section, its relationship with the archive at Zenodo itself is less clear. Specifically, the archive is comprised of a series of large, and anonymous, zipfile fragments, without even a README explaining the organisation. It would be more helpful to readers if the contents of the archive were organised into code, data, and scripts, such that they could be selectively downloaded. Apologies if I am misunderstanding the structure of your archive, but its current organisation is unnecessarily difficult to understand.
We updated the files in Zenodo and now the contents appear as Version 4. In total, the folder (archived) we uploaded has a 4+ GB size, and due to file size restrictions, we divided model_output

and model_experiment folders into multiple zip files. But Zenodo is able to show the contents of the folders with this version. We also included a separate README file which can be separately downloaded. It contains detailed information on the contents of each subfolders. We believe this approach gives the reader understanding of what each component contain and the ability to download them separately.

Please note that, due to working with multiple computers, there was an inconsistency in this revision, as such the figure captions in Section 5 appeared in the main text. We have deleted those, but they appear in the track changes. We have not modified anything to the text that was already approved by the referees and the editor.